# Automated analysis of long-term grooming behavior in *Drosophila* using a *k*-nearest neighbors classifier

Bing Qiao[1†], Chiyuan Li[1†], Victoria W Allen[2], Mimi Shirasu-Hiza[2], Sheyum Syed[1*]

[1]Department of Physics, University of Miami, Coral Gables, United States; [2]Department of Genetics and Development, Columbia University, New York, United States

**Abstract** Despite being pervasive, the control of programmed grooming is poorly understood. We addressed this gap by developing a high-throughput platform that allows long-term detection of grooming in *Drosophila melanogaster*. In our method, a *k*-nearest neighbors algorithm automatically classifies fly behavior and finds grooming events with over 90% accuracy in diverse genotypes. Our data show that flies spend ~13% of their waking time grooming, driven largely by two major internal programs. One of these programs regulates the timing of grooming and involves the core circadian clock components *cycle*, *clock*, and *period*. The second program regulates the duration of grooming and, while dependent on *cycle* and *clock*, appears to be independent of *period*. This emerging dual control model in which one program controls timing and another controls duration, resembles the two-process regulatory model of sleep. Together, our quantitative approach presents the opportunity for further dissection of mechanisms controlling long-term grooming in *Drosophila*.
DOI: https://doi.org/10.7554/eLife.34497.001

**\*For correspondence:**
s.syed@miami.edu

[†]These authors contributed equally to this work

**Competing interests:** The authors declare that no competing interests exist.

## Introduction

Grooming is broadly defined as a class of behaviors directed at the external surface of the body. Most animals spend considerable time grooming (*Mooring et al., 2004*; *Sachs, 1988*), and this near universality suggests that grooming likely fulfills an essential role for animals (*Spruijt et al., 1992*). Grooming assumes a variety of forms in different species—for instance, birds preen the oily substance produced by the preening gland from their feathers and skin, cats and dogs lick their fur, and flies sweep their body parts with their legs. Although in most cases the primary function of grooming is to maintain a clean body surface, different species-specific forms of grooming have roles in diverse functions such as thermoregulation, communication and social relationships (*Dawkins and Dawkins, 1976*; *Ferkin et al., 2001*; *Geist and Walther, 1974*; *McKenna, 1978*; *Patenaude and Bovet, 1984*; *Schino, 2001*; *Schino et al., 1988*; *Seyfarth, 1977*; *Spruijt et al., 1992*; *Thiessen et al., 1977*; *Walther, 1984*).

Many animal behaviors, such as locomotion, have been shown to be controlled by both external stimuli (stimulated behavior) and internal programs (programmed behavior). An example of stimulated locomotor activity is the abrupt evasive response triggered by the sudden appearance of a predator. In contrast, programmed locomotor activities, such as daily foraging for food, are essential to maintain vital functions of the organism (*Bergman et al., 2000*). Similar to locomotion, limited data from mammals suggest that grooming may be controlled by both external stimuli and internal programs (*Hart et al., 1992*; *Hawlena et al., 2008*; *Mooring and Samuel, 1998*). For example, stimulated grooming might be performed when the animal is excessively dirty or itchy, and programmed

**eLife digest** From birds that preen their feathers to dogs that lick their fur, many animals groom themselves. They do so to stay clean, but routine grooming also has a range of other uses, such as social communication or controlling body temperature. Despite its importance, grooming remains poorly understood; it is especially unclear how this behavior is regulated.

Fruit flies could be a good model to study grooming because they are often used in laboratories to look into the genetic and brain mechanisms that control behavior. Flies clean themselves by sweeping their legs over their wings and body, but little is known about how the insects groom 'naturally' over long periods of time. This is partly because scientists have had to recognize and classify grooming behavior by eye, which is highly time-consuming.

Here, Qiao, Li et al. have created a system to automatically detect grooming behavior in fruit flies over time. First, a camera records the movement of an individual insect. A computer then analyzes the images and picks out general features of the fly's movement that can help work out what the insect is doing. For example, if a fly is moving its limbs, but not the main part of its body, it is probably grooming itself. Qiao, Li et al. then borrowed an algorithm from an area of computer science known as 'machine learning' to teach the computer how to classify each fly's behavior automatically.

The new system successfully recognized grooming behavior in over 90% of cases, and it revealed that fruit flies spend about 13% of their waking life grooming. It also showed that grooming seems to be controlled by two potentially independent internal programs. One program is tied to the internal body clock of the fly, and regulates when the insect grooms during the day. The other commands how long the fly cleans itself, and balances the amount of time spent on grooming with other behaviors.

Cleaning oneself is not just important for animals to stay disease-free: it also reflects the general health state of an individual. For example, a loss of grooming is associated with sickness, old age, and, in humans, with mental illness. If scientists can understand how grooming is controlled at the brain and molecular levels, this may give an insight into how these mechanisms relate to diseases. The system created by Qiao, Li et al. could help to make such studies possible.
DOI: https://doi.org/10.7554/eLife.34497.002

grooming might be performed as a social ritual. Although grooming is a widely observed behavior, the basic mechanisms regulating grooming are still not well understood.

The fruit fly *Drosophila melanogaster* is an ideal model organism with which to dissect the fundamental mechanisms of grooming and its relationship to other behaviors. The fly is known to be a frequent groomer with a rich repertoire of behaviors and a sophisticated genetic toolkit developed to study them (*Connolly, 1968*; *Owald et al., 2015*). The study of *Drosophila* grooming can be traced back to the 1960s (*Connolly, 1968*; *Szebenyi, 1969*), and notable progress has since been made in studying grooming stimulated by the application of dust particles to the insect exterior (*Hampel et al., 2015*; *Seeds et al., 2014*). While most grooming studies thus far have focused on stimulated grooming, understanding the mechanisms responsible for programmed grooming will not only identify components distinct to each type of grooming but also inform us about how programmed grooming is prioritized with regard to other programmed behaviors such as locomotion, feeding, and sleep in the same organism.

A major hurdle in detecting programmed grooming in *Drosophila* is the lack of practical methodology. In many cases, fly grooming events are extracted by eye (*King et al., 2016*; *Phillis et al., 1993*; *Yanagawa et al., 2014*). Consequently, these data report only conspicuous behaviors within relatively short durations of observation. To improve resolution and accuracy, a number of sophisticated video-tracking methods have been recently developed for fly behavior (*Kain et al., 2013*; *Mendes et al., 2013*). These designs are not amenable to easy scale-up for tracking multiple individuals simultaneously. Moreover, while several of these methods are sufficient for short-term monitoring (*Branson et al., 2009*; *Kabra et al., 2013*), continuous multi-hour measurements and rapid, automated quantification methods are required to dissect long-term, unstimulated fly grooming relative to other daily behaviors like locomotion and sleep.

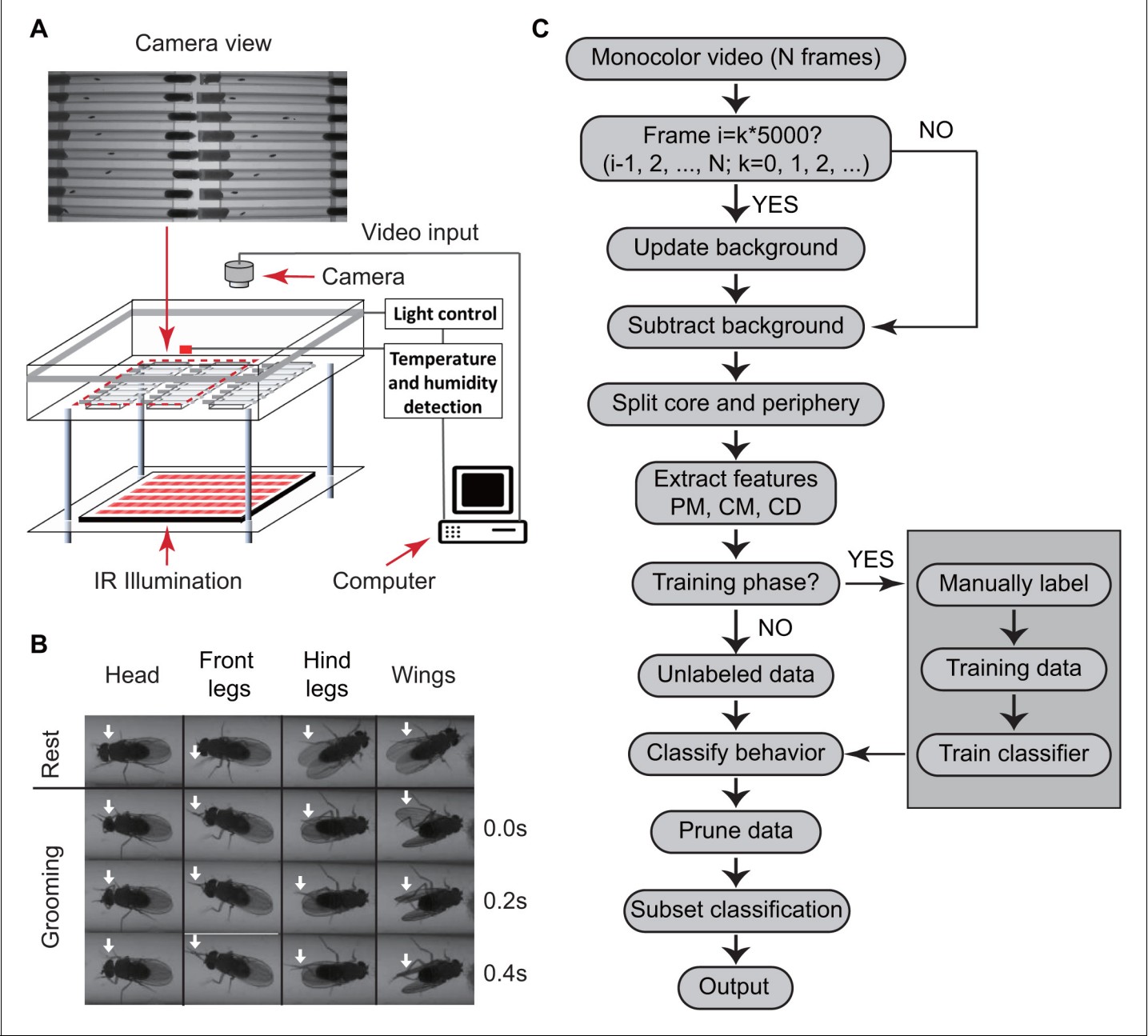

**Figure 1.** Overview of approach for detecting *Drosophila* grooming. (**A**) Apparatus used in recording behavior. Flies constrained to individual tubes are continuously illuminated by infrared light from below and recorded by a digital camera from above. LED lights on sides of chamber simulate day-night light conditions. Temperature and humidity probes placed in the chamber are monitored by a computer. Inset: Camera photo of fly tubes in chamber. (**B**) Examples of the most commonly observed types of grooming in our experiments. The top row displays postures of a fly in inactive state. The three rows below show how the limbs and body of a fly coordinate to perform specific grooming movements. Arrows point to the moving part during grooming. (**C**) Flowchart of our algorithm used to classify fly behavior. After generating a suitable background image, the algorithm characterizes movements of fly center (CD), core (CM) and periphery (PM) to fully classify behavior in each frame.

DOI: https://doi.org/10.7554/eLife.34497.003

To overcome limitations of currently available methods, we developed a new platform for long-term video-tracking and automated analysis of fly grooming. The layout of our hardware takes advantage of a basic design for housing individual flies that is widely used in locomotion and sleep studies (*Gilestro, 2012*; *Pfeiffenberger et al., 2010*; *Zimmerman et al., 2008*). Here, we

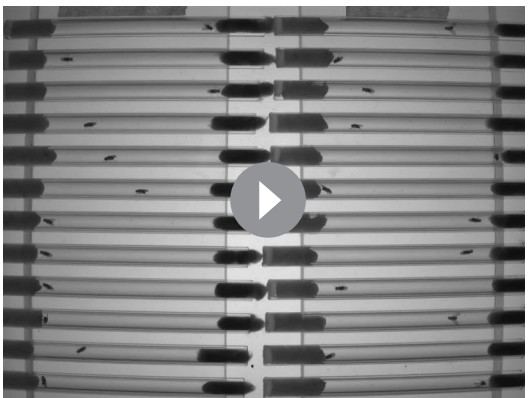

**Video 1.** Sample raw experimental video
DOI: https://doi.org/10.7554/eLife.34497.004

incorporate this standardized hardware into studies of grooming. Our algorithm maps fly activity onto a three-dimensional behavioral space and utilizes *k*-nearest neighbors (*k*NN) method, a machine learning technique, to classify each video frame as grooming, locomotion or rest. Results from multi-day recordings reveal that *Drosophila* spend approximately 13% of their waking time grooming, and the temporal pattern of grooming behavior is tightly regulated by the fly's internal circadian pacemaker. These findings suggest that grooming, similar to feeding and rest, likely serves one or more critical functions in *Drosophila*. Additionally, genetic perturbations reveal that the transcription factors CYCLE and CLOCK are critical parts of an internal program that controls the amount of *Drosophila* grooming. These grooming data, the easily implementable hardware, and the automated analysis package together permit the construction of high-resolution ethograms of stereotypical fly behavior over the circadian time-scale.

## Results

### Automatic grooming detecting system

We used a custom-designed video set-up to monitor fly behavior. Within the set-up, insects were placed individually in cylindrical glass tubes 6 cm long and 5 mm wide with food and cotton at opposite ends (*Figure 1A*). Tubes were placed in a chamber where temperature and humidity are monitored and controlled. Flies were illuminated from the sides by white light-emitting diodes (LED) to simulate day-night conditions and by infrared LED from below for video imaging. Videos were captured by a digital camera above the chambers (see Materials and methods). A sample raw video clip is shown in *Video 1*. Because the tubes (commonly used with Drosophila Activity Monitors or DAMs) are commercially available for studying circadian and sleep behavior, this set-up can be easily replicated by other labs.

We then developed an automated video image analysis package that classifies fly behavior into grooming, locomotion, or rest. 'Grooming' in our algorithm is defined as fly legs rubbing against each other or sweeping over the surface of the body and wings (*Szebenyi, 1969*) (*Videos 2* and

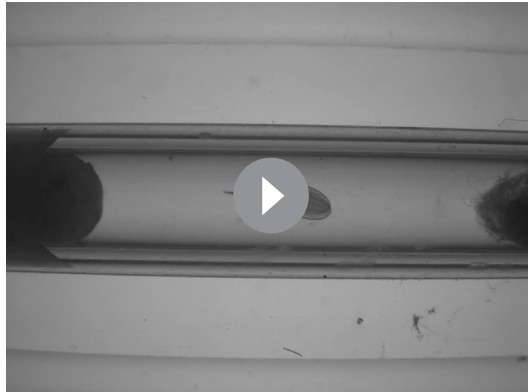

**Video 2.** Sample video of grooming on head and front legs
DOI: https://doi.org/10.7554/eLife.34497.005

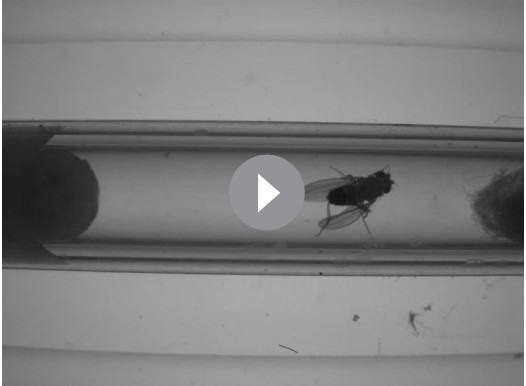

**Video 3.** Sample video of grooming on wings and hind legs
DOI: https://doi.org/10.7554/eLife.34497.006

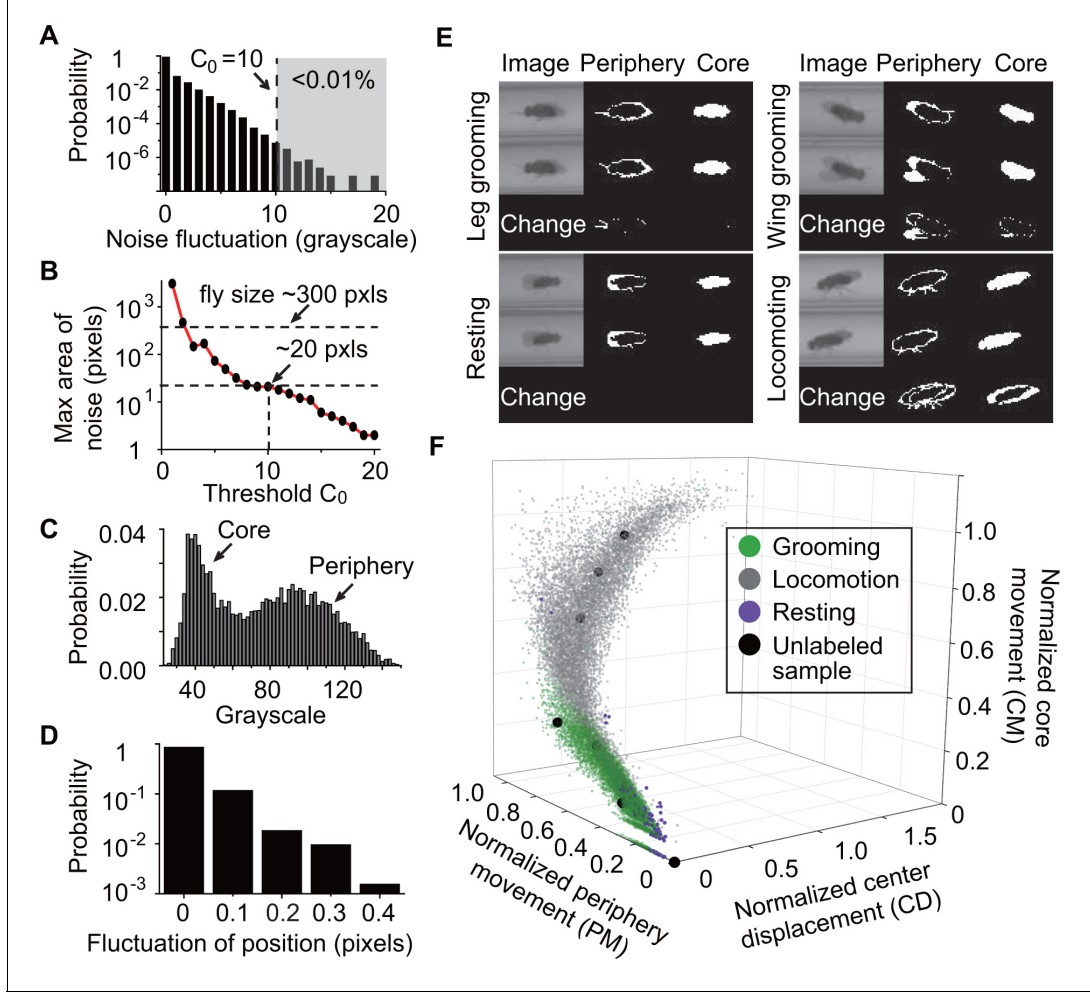

**Figure 2.** Feature extraction and behavior classification. (**A**) The distribution of grayscale fluctuations in the absence of mobile flies. A cutoff of grayscale value change $C_0 = 10$ rules out >99.99% of fluctuations. Shown here are only positive values of fluctuations, which are symmetric about zero. (**B**) Maximum area (pixels) of a closed object generated by noise when different threshold $C_0$ are applied. A $C_0 = 10$ rejects objects larger than 20 pixels. Based on this, we set a threshold $C_1 = 25$ to remove objects smaller than 25 pixels without affecting identification of flies which have a typical area of ~300 pixels in our studies. (**C**) Grayscale value distribution of pixels belonging to 20 individual flies. Two regions are clearly seen: the left region with peak around 40 represents the core of the flies and the right region with peak around 90 represents their periphery. (**D**) Variations in the center position of a stationary fly. The minimum displacement that represents a true fly center movement is 0.5-pixel length in our experiment, a requirement that excludes >99.99% of false displacements. (**E**) Examples of original and processed images of a fly displaying different behaviors: Top, left: front leg grooming; top, right: wing grooming; bottom, left: resting; bottom, right: locomoting. In each panel, original images from two consecutive frames are shown on left, periphery in the middle and core on the right. Changes of periphery and core are shown in the bottom row. PM and CM denote differences in the number of pixels representing the fly periphery and core, respectively, in two frames. Features PM and CM are different for different behaviors. Rubbing of front legs manifests through PM (top, left) while sweeping wings affects PM and CM (top, right). (**F**) k-nearest neighbors (kNN) algorithm works by placing an unclassified sample (black circle) representing a frame into a feature space with pre-labeled samples (green/gray/purple circles, the training set). The label of the unclassified point is decided by the most frequent label among its k-nearest neighbors. The three axes of the feature space are normalized periphery movement (PM), core movement (CM), and center displacement (CD). Fly activity in the feature space is separated into three regions: grooming (green), locomotion (gray) and resting (purple). Training samples (N = 9322 grooming, 9930 locomotion, 5748 rest) and nine unlabeled samples in PM-CM-CD space are shown.

DOI: https://doi.org/10.7554/eLife.34497.007

The following source data and figure supplement are available for figure 2:

**Source data 1.** Source data for *Figure 2*.
DOI: https://doi.org/10.7554/eLife.34497.009
**Source data 2.** Source data for *Figure 2—figure supplement 1*.
DOI: https://doi.org/10.7554/eLife.34497.010

**Figure supplement 1.** Details of environmental conditions and fly detection.

*Figure 2 continued*

DOI: https://doi.org/10.7554/eLife.34497.008

*3*), 'locomotion' as translation of the whole body, and 'rest' as the absence of either grooming or locomotion. *Figure 1B* shows images of grooming behaviors frequently observed in our videos involving the head, legs and wings. Since we are primarily interested in detecting grooming events rather than performing a detailed classification of all types of behavior (*Branson et al., 2009*), other behaviors involving body centroid movements, such as feeding, were initially classified as locomotion. This three-tier classification allowed our algorithm to efficiently and rapidly interpret grooming events in the recordings without incurring any significant errors in reporting locomotion and rest.

## Behavior classification algorithm

To classify behavior, raw videos were processed through four major automated steps: fly identification, feature extraction, classifier training (optional), and subset behavior classification (*Figure 1C*). First, fly identification was accomplished with the following analysis. Fly shape was extracted from a video frame by computing the difference between the current frame and a reference frame. The reference or background frame was created by comparing eight randomly selected frames and erasing all moving objects from one of them (see Materials and methods). The background frame was updated every 1000 s to account for changes in the fly's surroundings, such as decrease in the level of food and accumulation of debris within the tube, over the course of multiple hours (*Figure 2—figure supplement 1B*). A preliminary image of flies in the current frame was determined by comparing the frame to background and setting all pixels greater than a threshold $C_0$ (*Figure 2A*) equal to 10. Despite the use of $C_0$, some artifacts in the form of small objects still remained in the extracted image. A $C_0 = 10$ rejects artifacts larger than 20 pixels (*Figure 2B*). Based on this, to further eliminate remaining small objects, we erased all closed objects with areas less than a second threshold $C_1$ = 25 pixels, retaining only the fly silhouette (*Figure 2—figure supplement 1C*, right). Thus, each individual fly and its movements were distinguished from background structures.

Second, we performed feature extraction to distinguish three specific types of behaviors, which are grooming, locomotion, and rest, performed by the individual fly. The features we used were: (1) periphery movement (PM), which characterizes movements of the legs, head and wings; (2) core movement (CM), which quantifies movements of the thorax and abdomen; and (3) centroid displacement (CD), which quantifies whole body displacement. Extracting these three features allowed us to identify patterns corresponding to different types of behavior.

To extract PM and CM, we split each fly's body into a core and a periphery. Based on the grayscale distributions of the two parts (*Figure 2C*), we set the median of pixel grayscale values as the criterion to split a fly body into core (darker) and periphery (lighter). This criterion made the core and periphery areas roughly equal, giving PM and CM equal weight in the feature space. Slight variations in light condition across the arena can cause differing grayscale distribution for each individual. We therefore calculated the median value separately for each fly. After splitting the fly's body into two parts, PM and CM were extracted by computing the number of non-overlapping periphery and core pixels, respectively, in two consecutive frames.

To extract CD, we calculated the average position of all pixels from the individual fly and defined changes in that quantity between every two consecutive frames as CD. Since the fly moves in essentially one dimension through the narrow tube, we ignored movements perpendicular to the long axis of the tube when calculating centroid movement. In subsequent analysis, fly location was represented by its centroid position. Noise in the apparatus may slightly change the centroid position even when a fly is stationary. *Figure 2D* shows the distribution of such centroid displacements caused by noise. Based on this distribution, we set 0.5 pixel length as the minimum actual CD – that is, displacements smaller than 0.5 pixel were ignored. Application of this threshold eliminated 99.99% of such false displacements and accurately identified fly centroid displacement.

By extracting these three features (PM, CM and CD), we were able to distinguish between locomotion, rest, and grooming. As shown in *Figure 2E*, relative metrics of PM and CM were different depending on the type of behavior. Specifically, during locomotion, both parts moved significantly (*Figure 2E*, bottom-right) together with substantial changes in CD. During rest, no significant

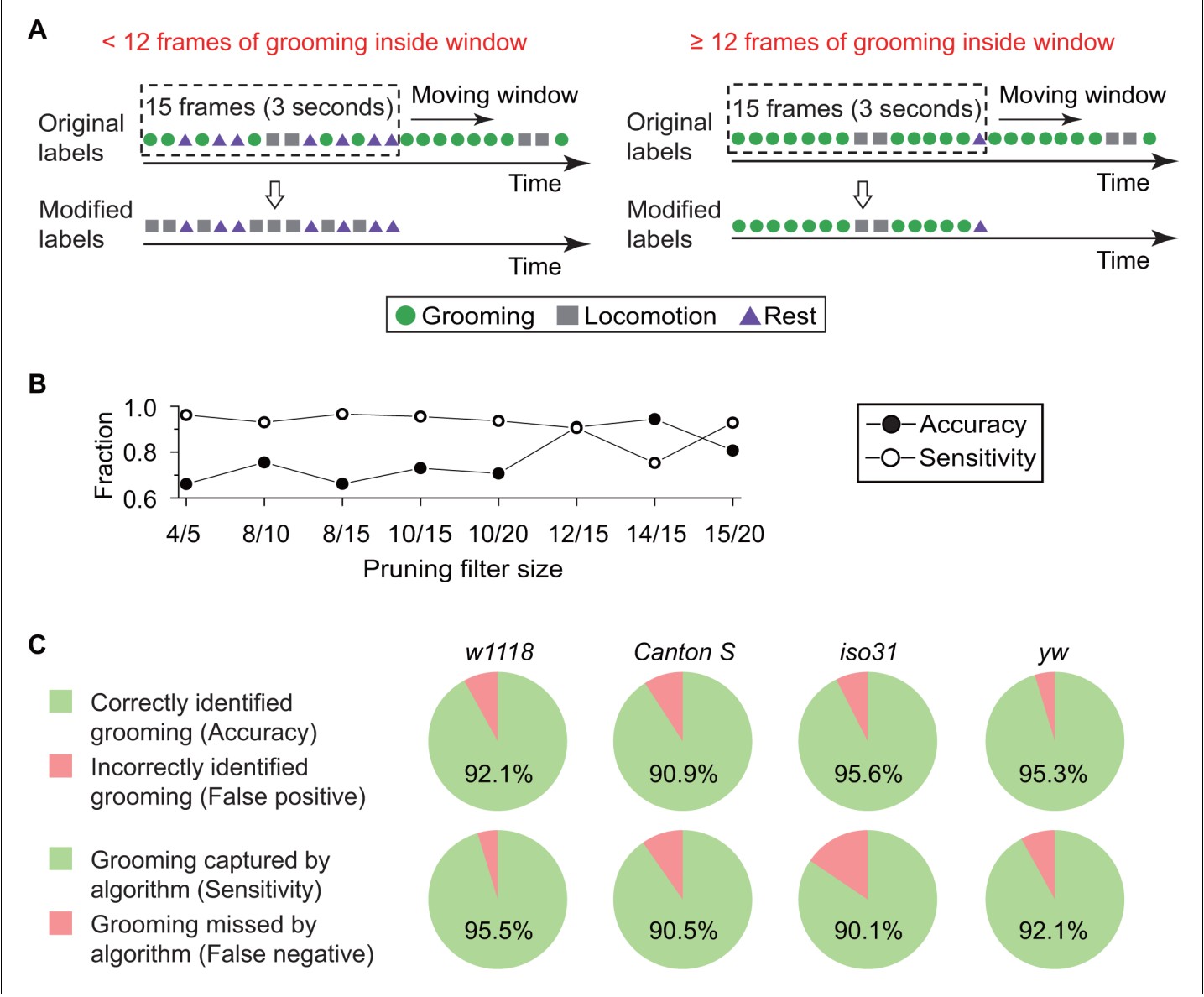

**Figure 3.** Data pruning and performance evaluation. (**A**) Grooming data are pruned after identification by the *k*NN classifier. A frame is finally labeled as grooming only if this frame is in a group of 15 frames in which 12 or more were labeled as grooming by the classifier (see B below). Frame previously labeled as grooming by the classifier but that did not pass the pruning procedure is relabeled as locomotion. (**B**) Performance of the classifier with pruning filter sizes of 4/5, 8/10, 8/15, 10/15, 10/20, 12/15, 14/15 and 15/20. Accuracy (closed circles) is equal to the ratio of correct grooming labels to all output grooming labels. Sensitivity (open circles) is equal to the ratio of grooming identified by the classifier to all visually labeled grooming events. We set the pruning filter to be 12/15 to attain >90% accuracy and sensitivity. (**C**) Fly genotypes vary by size and pigmentation, which can potentially affect performance of our classifier. To verify the generality and robustness of our method to different genotypes, accuracy (top) and sensitivity (bottom) of classifier on *w*[1118], *Canton S*, *iso31*, and *yw* were tested. Error rates in all tested strains were less than 10%.

DOI: https://doi.org/10.7554/eLife.34497.011

The following source data is available for figure 3:

**Source data 1.** Source data for *Figure 3*.
DOI: https://doi.org/10.7554/eLife.34497.012

movement was seen either in the periphery or the core (*Figure 2E*, bottom-left). During grooming, the periphery moved more than the core (*Figure 2E*, top-left, top-right). Importantly, since differences in fly size can affect values of PM, CM and CD, we normalized these features to individual fly size before proceeding with further analysis (see Materials and methods). The behavior-dependent

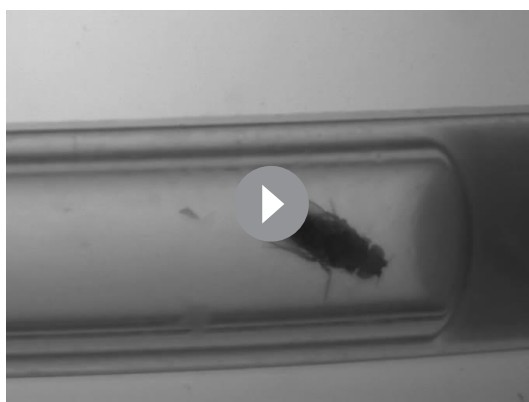

**Video 4.** Sample video of grooming-like behavior (stretching body)
DOI: https://doi.org/10.7554/eLife.34497.013

changes of these features suggest that PM, CM and CD are appropriate metrics for behavior classification.

Third, to produce a rapid, objective and auto-mated quantification of grooming behavior, we performed classifier training to teach the algo-rithm to automatically recognize these features. We classified fly behavior by applying the *k*-near-est neighbors (*k*NN) technique to the normalized features (*Bishop, 2007*; *Dankert et al., 2009*; *Kain et al., 2013*). Briefly, *k*NN works by placing an unlabeled sample into a feature space with pre-labeled samples serving as a training set for the algorithm. The label or class of the unlabeled sample is then decided by the label that is most common among its *k*-nearest training samples. In our case, the nearest neighbors were searched through a *k*-d tree algorithm (*Sproull, 1991*). To construct the *k*NN classifier, we prepared a train-ing set by visually labeling fly behavior from 25,000 frames (9322 frames of grooming, 9930 frames of locomotion and 5748 frames of resting from 20 $w^{1118}$ flies) and mapping them onto a three-dimensional feature space where the axes correspond to normalized PM, CM and CD (*Figure 2F*, color symbols). With these training samples, we applied 10-fold cross-validation (*Bishop, 2007*; *McLachlan et al., 2005*) to the *k*NN classifier with *k* ranging from 1 to 50 and settled on *k* = 10 to achieve balance between computing time and accuracy (*Figure 2—figure supplement 1D*).

Finally, to specifically distinguish between grooming behavior and other types of peripheral movement, we pruned output labels from the *k*NN classifier (*Figure 3A*). The algorithm calculates features from every two consecutive frames, resulting in some classifications being confounded by short-term fly activity. For example, features extracted from only two frames often cannot distinguish a fly stretching its body parts from one that is grooming (*Video 4*). Based on our observations during creation of the training set, a typical bout of grooming lasts >3 s or for 15 frames at our normal frame rate, longer than an average stretching event, which lasts for ~1 s. Accordingly, we devised a strategy in which a ~ 15 frame-long temporal filter slid one frame at a time to eliminate false groom-ing labels caused by short, grooming-like behavior. Grooming designations were retained only if at least a minimum number of grooming frames were found within the filter (*Figure 3A*). To determine the size of the filter and the minimum number of grooming frames within, we assessed the accuracy of our classifier with the 'minimum number of grooming frames/size of filter' at 4/5, 8/10, 8/15, 10/15, 10/20, 12/15, 14/15, and 15/20. These tests were conducted with a 10 min video (N = 20 *Canton S* flies). As expected, comparison between 8/15, 10/15, 12/15 and 14/15 shows (*Figure 3B*) that for fixed filter sizes, a larger number of grooming frames led to fewer false positive (higher accuracy) but more frequent false negative identification of grooming (lower sensitivity). On the other hand, <12 minimum number increased risk of misidentifying other short-term grooming-like behav-iors as grooming. Based on these findings, we set the pruning filter to be 12/15, simultaneously min-imizing false positive and false negative errors. Because of this pruning process, if fewer than 12 grooming frames were found within a 15-frame sliding window, then all grooming frames were re-labeled as locomotion once the left edge of the window reached the 15th frame (*Figure 3A*). Thus, these pruned labels were the final output of our grooming classification algorithm, consisting of fly identification, feature extraction and classifier training.

The accuracy of our algorithm was evaluated by comparing the computer-identified grooming with manually labeled grooming identified by visual inspection. We tested a total of 450 min of vid-eos from a different set of $w^{1118}$ flies (N = 15) than the one used in training the classifier. The com-parisons showed that, of the grooming events picked out by our algorithm, 92.1% were manually verified as true grooming events (*Figure 3C*, top panel). Furthermore, among all manually scored grooming events, 95.5% were successfully identified by our computational method (*Figure 3C*, bot-tom panel). Since size and pigmentation differences between genotypes can potentially affect behavioral classification, we investigated robustness of our $w^{1118}$-trained classifier with manually-

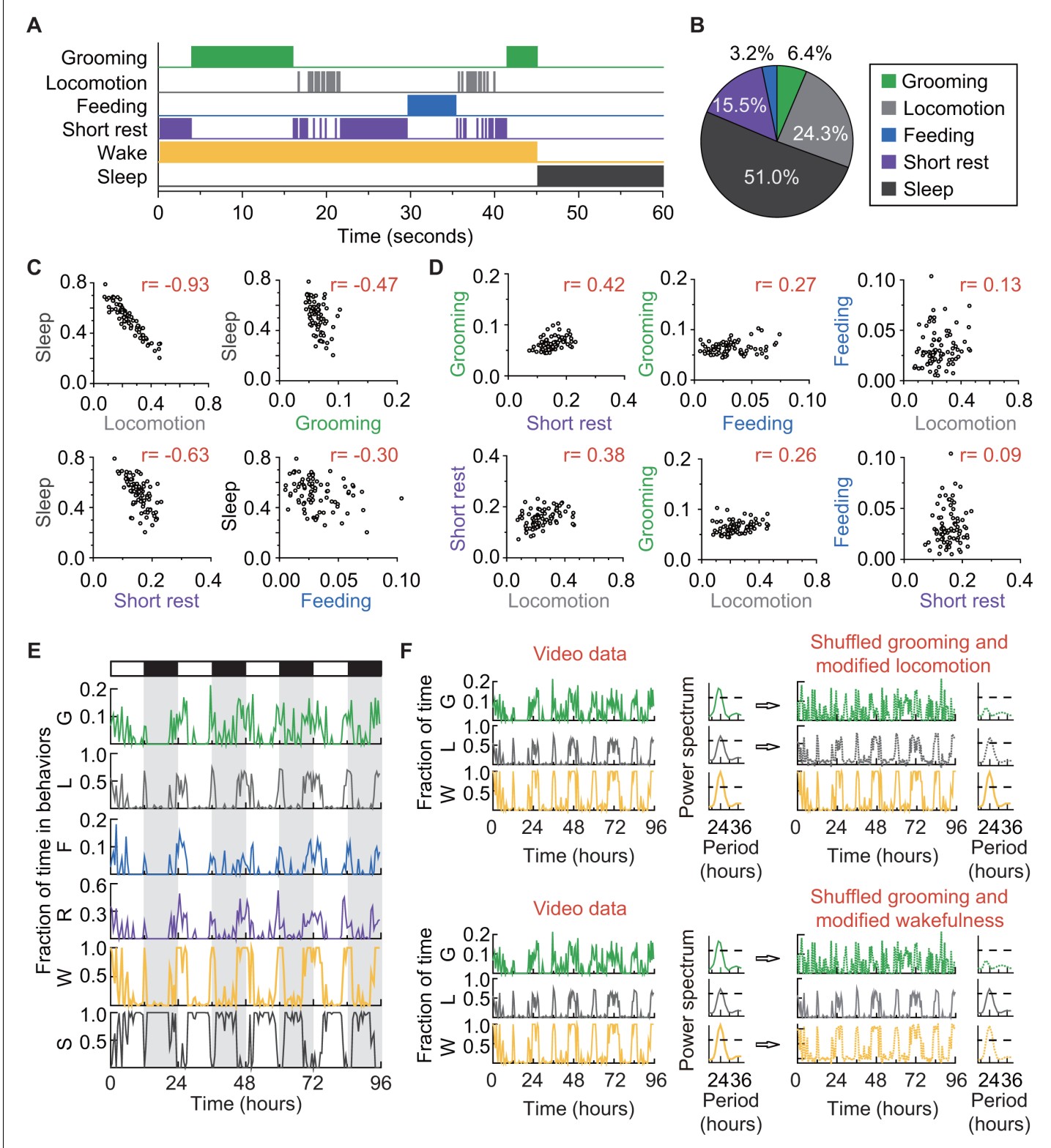

**Figure 4.** How grooming fits into the daily routine of a fly. (**A**) Ethogram of grooming (green), locomotion (gray), feeding (blue), short rest (purple), and sleep (dark gray) performed by an *iso31+* fly in 60 s (300 frames). Individual events of these four behaviors are mutually exclusive and together constitute wake (yellow-orange), which is complementary to sleep (dark gray). (**B**) Average fraction of time flies spent in each behavior. N = 83 *iso31+* flies. (**C**) (**D**) Correlation between pairs of behaviors. There is strong negative correlation between sleep and locomotion (r = −0.93) and between sleep and short rest (r = −0.63). Interestingly, time spent in grooming does not show strong correlation with any of the other four behaviors. N = 83 *iso31+*

*Figure 4 continued on next page*

*Figure 4 continued*

flies. r is the Pearson product-moment correlation coefficient. (**E**) Temporal patterns of behaviors of a single *iso31+* fly during 4 days in LD cycles. Behaviors shown here are, grooming (**G**), locomotion (**L**), feeding (**F**), short rest (**R**), wake (**W**), and sleep (**S**). Level of activity is shown in terms of fraction of time spent in each behavior. Fraction is calculated every 30 min. White/black horizontal bars indicate light/dark environmental conditions, respectively. (**F**) Rhythmicity in grooming, locomotion and wake in an example fly. In LD condition, fraction of time spent in these behaviors are plotted on left. In power spectra on right of time series of behaviors (horizontal dash line denotes threshold power for p=0.05), temporal patterns of the three behaviors all show significant circadian rhythmicity. In right top, spectra of randomized grooming show no rhythmicity, while modified locomotion is still rhythmic. Similarly, in time series on right bottom, with the same randomized grooming, wake remains rhythmic while grooming, as one component from it, is arrhythmic. In time series of behaviors, activity is binned every 30 min.

DOI: https://doi.org/10.7554/eLife.34497.014

The following source data and figure supplements are available for figure 4:

**Source data 1.** Source data for *Figure 4*.
DOI: https://doi.org/10.7554/eLife.34497.018
**Source data 2.** Source data for *Figure 4—figure supplement 1*.
DOI: https://doi.org/10.7554/eLife.34497.019
**Source data 3.** Source data for *Figure 4—figure supplement 2*.
DOI: https://doi.org/10.7554/eLife.34497.020
**Figure supplement 1.** Relationships among fly grooming, locomotion, feeding, short rest, and sleep.
DOI: https://doi.org/10.7554/eLife.34497.015
**Figure supplement 2.** Temporal relationships between grooming and locomotion.
DOI: https://doi.org/10.7554/eLife.34497.016
**Figure supplement 3.** Mathematical description of temporal changes in grooming and locomotion patterns.
DOI: https://doi.org/10.7554/eLife.34497.017

labeled data from *Canton S*, *iso31*, and *yw* strains (10 min videos with N = 20 of each type). As shown in *Figure 3C*, error rates in each tested strain less than 10%. Together, these results suggest that our method identifies grooming with high fidelity in several different *Drosophila melanogaster* strains.

## Flies spend a significant portion of their awake time grooming

The solitary flies in our experiments also spent portions of their time feeding (*Ja et al., 2007*) and sleeping (*Hendricks et al., 2000*; *Shaw et al., 2000*), behaviors that our classifier did not initially label but that can nevertheless be identified by our algorithm. Prolonged proximity with food (>3 s, <body length) was accepted as a proxy for feeding. Rest periods lasting ≥5 min (*Dubowy and Sehgal, 2017*) were classified as sleep, following the currently accepted definition of the behavior. Together, these additional classifications led to the identification of five major behaviors in our data: grooming, locomotion, feeding, short rest (< 5 min of quiescence), and sleep (*Figure 4*). The first four behaviors are mutually exclusive at the level of single events, together defining the wake state of the fly, and collectively complementary to the sleep state (*Figure 4A*). We found that a typical *iso31⁺* fly under 12 hr light:12 hr dark (LD) conditions spent approximately 6% of its daily time grooming,~24% time locomoting, ~3% time feeding, ~16% resting, and the remaining ~51% sleeping (*Figure 4B*). That is, the average *iso31⁺* fly spent ~13% of its awake time grooming. It is worth noting here that such behavioral statistics can vary even between wild-type laboratory strains (*Colomb and Brembs, 2014*; *Zalucki et al., 2015*). For instance, similar analysis of a *Canton-S* strain showed that these flies groomed ~19% of their awake time (*Figure 4—figure supplement 1A*). These analyses demonstrate that our platform for long-term video-tracking and automated analysis can provide a quantitative ethological structure for daily basal fly behavior.

Since sleep and wake are complementary states, we expected fractional time spent in sleep to negatively correlate with that of the four wake behaviors our method tracks. Pair-wise comparisons (Pearson's correlation coefficient, r, see Materials and methods) of individual fly sleep with grooming, locomotion, short rest or feeding, showed the expected negative relationships (*Figure 4C* and *Figure 4—figure supplement 1B*). Interestingly, the strength of negative correlation with sleep (*Figure 4C*) increased with the average fractional time spent in a wake behavior (*Figure 4B*). We reasoned that similar analysis among the wake behaviors, in contrast, should show positive correlations. Pair-wise comparisons among grooming, locomotion, short rest and feeding showed the predicted

positive correlations, although to varying degrees (*Figure 4D* and *Figure 4—figure supplement 1C*). The analyses further revealed that the fraction of time a fly spent in short rest was the best predictor of its grooming time (r = 0.42 in *iso31*+ and 0.26 in *Canton-S*) while locomotion (r = 0.26 and −0.13) and feeding (r = 0.27 and 0.06) were both less reliable in predicting grooming.

The weaker grooming-locomotion and grooming-feeding correlations were unexpected for two reasons. First, daily variations in grooming levels had appeared to closely follow those in locomotion (*Figure 4—figure supplement 2A*), suggesting the possibility that grooming is a by-product of the more robustly driven locomotor activity. Second, feeding activity has been postulated to act as a trigger for grooming with food debris serving as an external stimulus (*Hampel et al., 2015*; *Seeds et al., 2014*). To further dissect the lack of predictive relationship between grooming and locomotion, we first examined temporal parameters that describe grooming and locomotion over short timescales (*Figure 4—figure supplement 2A–E*). Basal locomotor events during mid-day and night (*Figure 4—figure supplement 2A*, rectangles) were relatively sparse compared to grooming episodes during the same times. This difference in inter-event time interval between grooming and locomotion persisted to different degrees throughout the day-night cycle, such that the average longest pause between two subsequent grooming events was ~88 min while that between two locomotor events was ~132 min (*Figure 4—figure supplement 2C*). Examination of the duration of individual events showed grooming events on average lasted for ~0.23 min compared to ~0.44 for locomotor events (*Figure 4—figure supplement 2D*). These analyses revealed significant differences between the two behaviors over short timescales and do not support locomotor activity as a driver of grooming.

To focus on temporal dynamics at longer timescales, we binned multi-day data in 30 min (*Figure 4—figure supplement 2F,G*) and applied least square fit to a previously developed mathematical model that describes long timescale variations in fly activity in terms of exponential functions (*Lazopulo and Syed, 2016*, *2017*). The functions were defined by four rate parameters $b_{MR}$, $b_{MD}$, $b_{ER}$ and $b_{ED}$, where subscripts denote morning rise (MR), morning decay (MD), evening rise (ER) and evening decay (ED), and two duration parameters that describe the relative durations of morning (TM) and evening (TE) peaks in activity (*Figure 4—figure supplement 2H* and *Figure 4—figure supplement 3*). Results from this analysis showed that the rate parameter $b_{MR}$ of grooming was smaller than that of locomotion (*Figure 4—figure supplement 2I*), indicating a slower increase in night-time grooming activity. Additionally, the evening duration parameter (TE) for grooming was greater than that for locomotion (*Figure 4—figure supplement 2J*), indicating that the evening peak in grooming lasted longer. These differences in long timescale kinetics were again inconsistent with locomotor activity as a driver of grooming. Finally, comparison with large timescale variations in feeding patterns showed that peak time in contacting food was offset by 2–4 hr from nearby peaks in grooming (*Figure 4—figure supplement 2O–P*). The large temporal offset suggests contact with food is also not likely to drive the majority of grooming events observed in our experiments. Thus, according to our analyses of the kinetics of *Drosophila* ethograms in our system, neither locomotor activity nor feeding is likely to be a primary driver of basal grooming.

To identify major drivers of basal grooming, we noted that multi-day time series of the behaviors showed time-of-day-dependent changes in each behavior (*Figure 4E*). The appearance of repeating patterns raised the possibility that external light-dark (LD) cycles alone or in combination with internal programs could be exerting temporal control over several of these behavioral outputs, including grooming. Indeed, environmental light-dark cycles through influence on the circadian clock are known to drive rhythmic changes in fly sleep and wake durations and within the awake state, feeding, and locomotor activities (*Chatterjee and Rouyer, 2016*; *Pfeiffenberger et al., 2010*). That these rhythms persisted in the absence of LD cycles is generally considered to be strong support for clock control of these behaviors.

We set out to determine whether the circadian clock drives rhythmic modulations in fly daily grooming independent of other circadian-regulated behaviors–that is, to test whether grooming exhibits circadian oscillations simply because individual grooming events are mutually exclusive of other individual wake activities. We recognized that the mutual exclusivity of the behaviors seen at the level of individual events (*Figure 4A*) did not persist at the level of fractional time in each behavior where the long timescale modulations are visible (*Figure 4E*). This is because fractional time data are binned and the only constraint on these data was that the sum of the time spent in each wake behavior (grooming, locomotion, feeding and short rest) and sleep equaled one for each time bin

(*Figure 4—figure supplement 1F*). In this representation, therefore, rhythmicity of one behavior (i.e. grooming) did not dictate rhythmic status of another (i.e. locomotion).

To test the independence of rhythms, we performed a series of 'shuffling experiments' using well-established (*Allada and Chung, 2010*; *Chatterjee and Rouyer, 2016*) rhythmicities of wakefulness and locomotion as metrics (*Figure 4F*). In brief, we took data from *Figure 4E* in which grooming, locomotion and wakefulness have LD-driven ~24 hr rhythms (*Figure 4F*, left and power spectra) and computationally randomized the grooming time-series such that it lost rhythmicity (*Figure 4F*, right). To account for the randomized grooming, we also adjusted either locomotion (*Figure 4F*, upper panel) or wakefulness (*Figure 4F*, bottom panel), in both cases ensuring that wakefulness was between 0 and 1 at all times (see Materials and methods). In either case, we found that rhythmicity in locomotion and wakefulness were intact regardless of the rhythmic status of grooming. The simulation result suggested that circadian control of fly locomotion and wakefulness does not guarantee circadian control of underlying basal grooming, at least as measured from changes in the duration of the behaviors. Therefore, demonstration of robust ~24 hr rhythms in grooming in the absence of any external cues should be strong evidence in favor of circadian control of the behavior.

## Temporal pattern of grooming is controlled by the circadian clock

To test whether basal grooming is also under circadian control, we first entrained $iso31^+$ + to 2 days of alternating light-dark cycles and then monitored their behavior over multiple days in constant darkness (DD). In the absence of light cues, locomotor, feeding and sleep showed the familiar clock-driven rhythms in their daily timing (*Figure 5A,B*). Although short rest appeared to undergo rhythmic changes (*Figure 5A*), spectral analysis indicated these changes did not result in statistically significant rhythms at the p=0.05 level (*Figure 5B*). Lack of rhythms in short rest is consistent with our earlier reasoning that rhythmic wakefulness and locomotion does not necessarily imply rhythmicity of each behavior in the awake state.

Grooming data also showed periodic changes in constant darkness (*Figure 5C*). Power spectra of individual time-series ('WT' in *Figure 5D* and *Figure 5—figure supplement 1A*) indicated these periodic changes to be statistically rhythmic by revealing peaks significant at p=0.01 in 100% of flies (29 out of 29 individuals, *Figure 5E*). The average period of oscillations was 23.72 hr, with a standard deviation of 0.72 hr (*Figure 5—figure supplement 1B*). The presence of these robust circadian rhythms in the absence of external cues further support the hypothesis that fly basal grooming is under control of the internal timekeeper. Consistent with our prediction that grooming rhythms in DD do not necessarily follow from rhythms in locomotion or wakefulness, we found that knowing locomotion or wakefulness is rhythmic did not inform about the rhythmic status of grooming (*Figure 5—figure supplement 2*). This finding further underscored the importance of the DD studies in establishing rhythmicity in basal grooming. It should be noted here that our simulation results do not demonstrate bidirectional independence of rhythmicity in wakefulness and grooming but, only that rhythmicity of wakefulness does not depend on that of grooming. Demonstration of fully independent rhythms in the two behaviors is beyond the scope of the present study.

We next took advantage of several circadian mutants to examine further the control of grooming by the circadian clock. The *Drosophila* clock is composed of two interlocked genetic feedback loops in which *period* (*per*) is one of the core components and whose transcription is controlled by the primary transcription factors Clock (clk) and Cycle (cyc) (*Allada and Chung, 2010*). The *per* gene has several well-characterized mutant alleles, two of which—$per^S$ and $per^L$—produce short and long circadian rhythms, respectively, while a third, $per^0$, results in arrhythmic behavior (*Konopka and Benzer, 1971*). Population-averaged grooming of $per^S$ and $per^L$ showed altered oscillations in LD and DD (*Figure 5C*, second and third rows), with average DD periods of 19.23 $\pm$ 0.57 hr and 28.84 $\pm$ 1.13 hr, respectively (*Figure 5D,E* and *Figure 5—figure supplement 1A*). The periods of oscillation in grooming were well within published values of circadian rhythms of these mutants (*Konopka and Benzer, 1971*) and in agreement with alterations in locomotor rhythms of the flies (*Figure 5—figure supplement 3A*). Consistent with these results, grooming in $per^0$ flies was arrhythmic (*Figure 5C*, bottom row) and, when analyzed at the individual fly level, the power spectra unveiled the absence of statistically significant rhythms in 19 out of 20 flies at p=0.01 level (*Figure 5D,E* and *Figure 5—figure supplement 1A*). Moreover, analysis of grooming patterns in $cyc^{01}$ (*Rutila et al., 1998*) and $clk^{Jrk}$ (*Allada et al., 1998*), arrhythmic mutants of *cyc* and *clk*, also

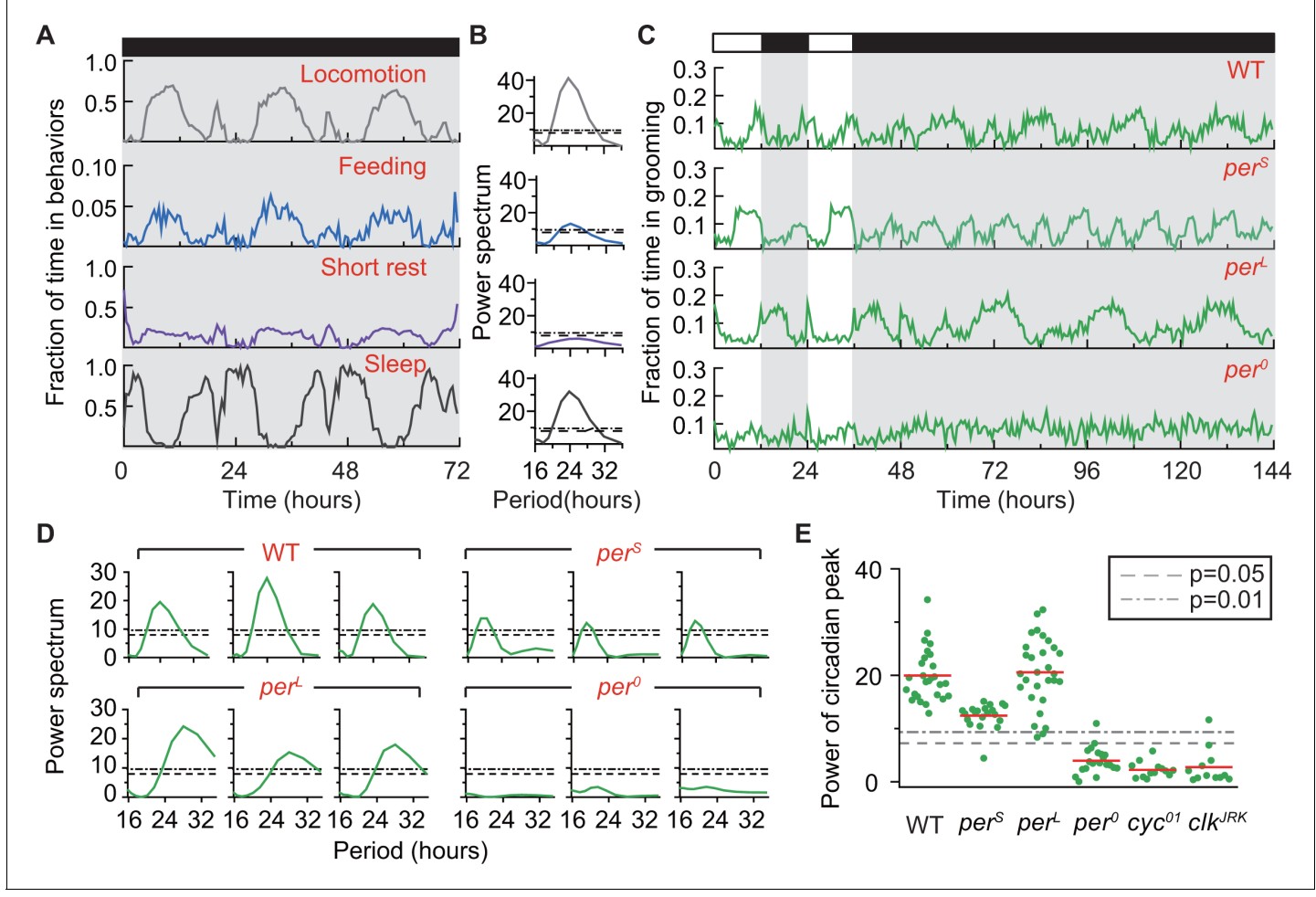

**Figure 5.** Grooming is under control of the circadian clock. (**A**) Average temporal patterns (fraction of time spent in 30 min bins) of locomotion, feeding, short rest and sleep of eight representative *iso31+* flies during 3 days in constant darkness (DD). Black horizontal bar represents lights-off condition. (**B**) Power spectra of behaviors in panel (**A**). Except for short rest, temporal patterns of the other three behaviors show significant circadian rhythmicity. Horizontal dash line and dash dot line denote threshold powers for p=0.05 and p=0.01, respectively. (**C**) Grooming activity (in 30 min bins) of wild-type and clock mutants during 2 days in LD cycle followed by four days in DD cycle. Grooming traces are population averages. In DD, wild-type (WT, *iso31+*) grooming continues to show 24 hr rhythms. In comparison, grooming in *perS* or *perL* flies show shorter or longer rhythms, respectively. For *per0* flies, grooming is arrhythmic in DD. N = 8 WT, 8 *perS*, 8 *perL*, and 8 *per0* representative flies. (**D**) Example power spectra showing circadian rhythmicity in grooming patterns of three individual wild-type, *perS*, *perL* and *per0* flies. Spectra are normalized to variance of activity (in 30 min bins). Dash lines and dash dot lines represent threshold power at p=0.05 and p=0.01, respectively. More examples of individual power spectra are provided in *Figure 5—figure supplement 1*. (**E**) Spectral powers of circadian peaks of individual wild-type and circadian mutants. N = 29 control, 20 *perS*, 29 *perL*, 20 *per0*, 13 *cyc01* and 11 *clkJRK*.

DOI: https://doi.org/10.7554/eLife.34497.023

The following source data and figure supplements are available for figure 5:

**Source data 1.** Source data for *Figure 5*.
DOI: https://doi.org/10.7554/eLife.34497.027
**Source data 2.** Source data for *Figure 5—figure supplement 1*.
DOI: https://doi.org/10.7554/eLife.34497.028
**Source data 3.** Source data for *Figure 5—figure supplement 3*.
DOI: https://doi.org/10.7554/eLife.34497.029
**Source data 4.** Source data for *Figure 5—figure supplement 2*.
DOI: https://doi.org/10.7554/eLife.34497.030
**Figure supplement 1.** (A) Example Lomb-Scargle periodograms of grooming activity of individual *per* mutants and their background control (WT).
DOI: https://doi.org/10.7554/eLife.34497.024
**Figure supplement 2.** Rhythmicity in grooming patterns need not be a direct result of rhythmicity in locomotion or sleep-wake cycles.

*Figure 5 continued on next page*

*Figure 5 continued*

DOI: https://doi.org/10.7554/eLife.34497.025

**Figure supplement 3.** (A) Locomotion (in 30 min bins) of wild-type (*iso31+*) and clock mutants during two days in LD cycle followed by four days in DD cycle.

DOI: https://doi.org/10.7554/eLife.34497.026

showed loss of circadian rhythms (*Figure 5E* and *Figure 5—figure supplement 3B–D*). Together, these results support the hypothesis that the circadian clock temporally modulates fly grooming.

## Grooming duration is controlled by *cycle* and *clock*

To test whether, in addition to regulating the timing of grooming, the circadian clock also regulates grooming duration, we examined the average duration of grooming in circadian mutants. Despite causing major changes in temporal patterns of grooming, the *per$^0$* mutation did not significantly change the average duration of grooming in these flies (*Figure 6A*). In contrast, *cyc$^{01}$* and *clk$^{Jrk}$* mutants both exhibited increased daily average grooming relative to their respective genetic controls (*Figure 6B,C*). While both mutants exhibited increased grooming duration, this change was accompanied by opposing changes in their locomotion: *cyc$^{01}$* flies spent less time and *clk$^{Jrk}$* flies spent almost twice as much time in locomotion (*Figure 6B,C*, pie plots). Thus, the increase in *cyc$^{01}$* grooming came almost entirely from loss of locomotor activity while the increase in *clk$^{Jrk}$* grooming came from loss of sleep. These results support the hypothesis that locomotion and grooming are partly independent behaviors and further suggests that the *cyc$^{01}$* and *clk$^{Jrk}$* mutations alter the insect's internal homeostasis in distinct ways, similar to phenotypic differences reported previously in sleep studies involving *cyc$^{01}$* and *clk$^{Jrk}$* (*Hendricks et al., 2003*; *Shaw et al., 2002*). Importantly, together with *per$^0$* data, the results raise the possibility of non-circadian roles for *cyc* and *clk* in setting the duration of internally driven grooming in *Drosophila*.

*cycle* and *clock* have also been implicated in stress response, particularly in regulating level of sleep in response to sleep deprivation and adjusting locomotor output in response to nutrient unavailability (*Hendricks et al., 2003*; *Keene et al., 2010*; *Shaw et al., 2002*). Because grooming and sleep have both been previously linked to stress, we asked whether reduction in sleep is always accompanied by an increase in grooming as seen in our *clk$^{Jrk}$* data. To address this question, we examined relationship between grooming and sleep in standard LD cycles in two short-sleeping mutants–*fumin* and *sleepless* (*sss*). Consistent with the original studies (*Koh et al., 2008*; *Kume et al., 2005*), our method found both strains to have extremely low levels of sleep (*Figure 6D,E*, pie plots). But, while loss of sleep in *fumin* was accompanied by an upregulation in grooming (*Figure 6D*), loss of sleep in *sss* was accompanied by a dramatic downregulation in grooming, compared to control flies (*Figure 6E*). These divergent relationships between sleep and grooming (e.g. *sss* vs. *fumin*) and between locomotion and grooming (e.g. *clk$^{Jrk}$* vs. *cyc$^{01}$*) became more evident when individuals of different genotypes were compared together (*Figure 6—figure supplement 1F,G*). To better visualize the effects of disparate mutations, data of each genotype in these plots were normalized to the population-mean of its genetic control. These results suggest that resetting of the level of internally-driven grooming can occur via a number of ways with complex compensatory changes in sleep and locomotor behavior.

Accumulated data from our experiments suggest that grooming is an innate fly behavior controlled by two major regulators. One of these regulators controls temporal patterns in grooming and the other controls amount of time spent in grooming. Circadian genes *per*, *cyc* and *clk* are involved in controlling the timing of peaks/troughs in grooming rhythms while *cyc* and *clk* are also involved in setting how much time is spent grooming. The apparent absence of *per* from the second regulatory mechanism is consistent with the possibility that the two control mechanisms operate independently.

Nearly all animals tested exhibit daily basal grooming behavior, suggesting that grooming is not only fundamental to health but also reflects a generally healthy state. Consistent with this, loss of grooming is indicative of sickness behavior (*Hart, 1988*) associated with infection or old age, and, in the case of humans, mental illness. A greater understanding of the molecular mechanisms regulating grooming would provide insight into the principles and neural circuits underlying other complex programmed behaviors, as well as potentially identify biomarkers of pathological disease states. Critical

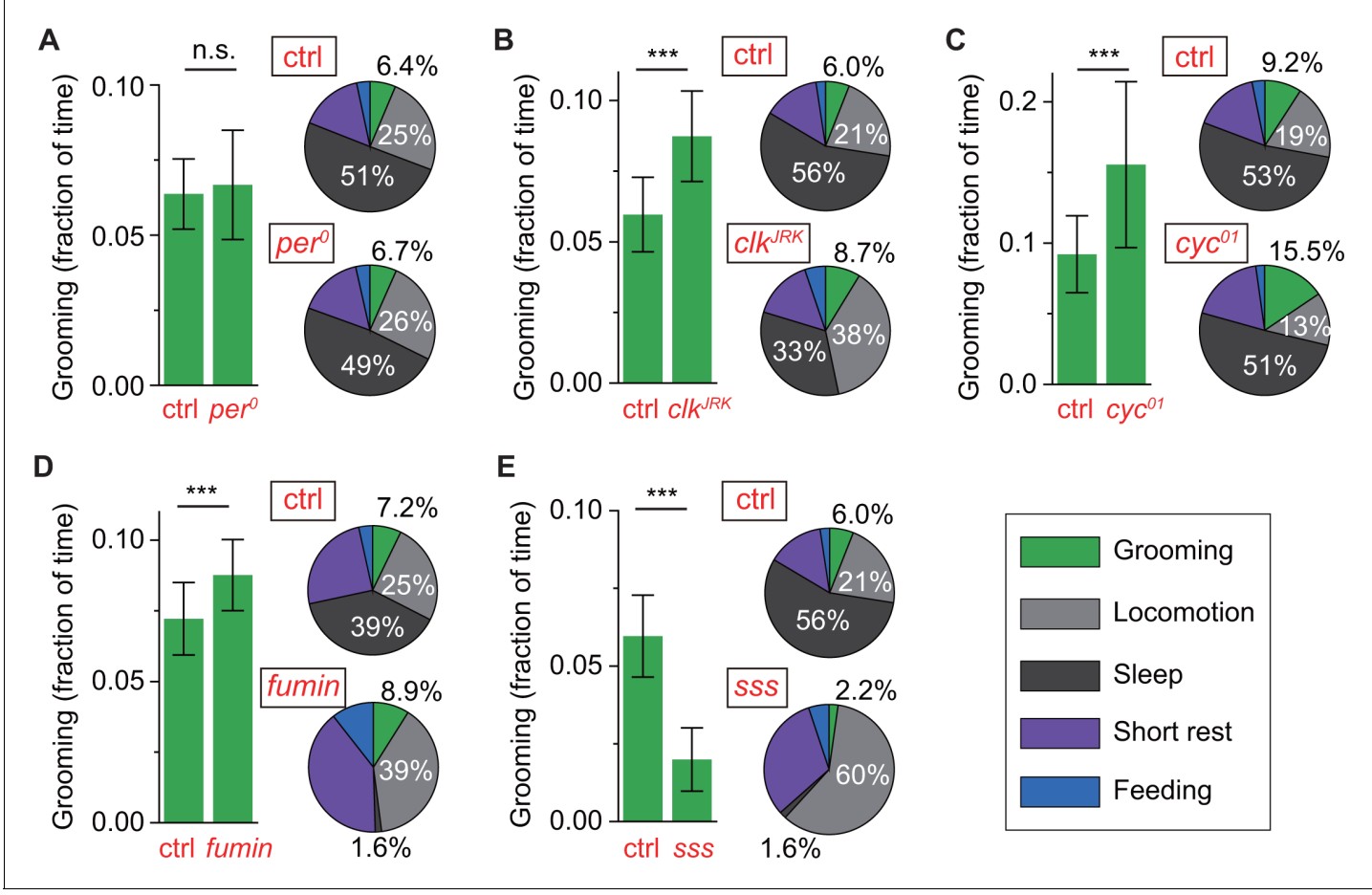

**Figure 6.** Control of grooming duration is independent of circadian rhythmicity. In each panel, bar plots on left show average fractional time spent in grooming in mutant and control flies. Pie charts on right present average fractional time spent in grooming (green), locomotion (gray), sleep (dark gray), short rest (purple) and feeding (blue). Here, numerical values for fractional time spent in behavior are indicated only for grooming, locomotion and sleep with additional details in *Figure 6—figure supplement 1A*. Although loss of a functional clock does not affect grooming amount (**A**), mutations in *clock* (**B**) and *cycle* (**C**) genes lead to robust increases in the time flies spend grooming. Additional time for grooming can come from reduction in sleep (**B**) or reduction in locomotion (**C**). Reduction in sleep, however, does not always entail similar changes in grooming since sleep mutants *fumin* (**D**) and *sleepless* (**E**) show divergent alterations in grooming durations. N = 83 control, 53 $per^0$, p=0.28. N = 76 control, 18 $cyc^{01}$, p=2.7×10$^{-4}$. N = 28 control, 25 $clk^{JRK}$, p=7.8×10$^{-9}$. N = 17 control, 23 *fumin*, p=0.003. N = 28 control, 17 *sss*, p=1.3×10$^{-10}$.

DOI: https://doi.org/10.7554/eLife.34497.031

The following source data and figure supplement are available for figure 6:

**Source data 1.** Source data for *Figure 6*.
DOI: https://doi.org/10.7554/eLife.34497.033
**Source data 2.** Source data for *Figure 6—figure supplement 1*.
DOI: https://doi.org/10.7554/eLife.34497.034
**Figure supplement 1.** Changes in grooming due to mutations in clock, sleep or immune genes.
DOI: https://doi.org/10.7554/eLife.34497.032

to the dissection of these molecular mechanisms is a system for rapid, automated interpretation of grooming in a genetically tractable model organism. The development of our platform will facilitate high-throughput and unbiased analysis of the genetic regulators and neural circuits that control grooming, as well as those responsible for loss of grooming in the context of disease.

## Discussion

Grooming continues to be one of the least understood *Drosophila* behaviors, possibly due to the technical challenges of detecting grooming events in this small insect. Early work describing fly grooming relied on manual scoring (*Connolly, 1968*; *Szebenyi, 1969*; *Tinbergen, 1965*), which imposes significant limitations on the length of events that can be detected, fidelity and objectivity of detection, and the level of detail that can be extracted from the data. Despite such limitations, these initial studies made a number of noteworthy observations. Szebenyi delineated all the major modes of fly grooming and suggested that repetitive grooming actions may closely follow a preset sequence (*Szebenyi, 1969*). A subsequent study in the blowfly offered a more refined mechanistic picture of insect grooming by proposing that the sequential actions form a hierarchical structure (*Dawkins and Dawkins, 1976*). Combining modern computational and genetic tools, an elegant study in *Drosophila* recently confirmed these previous hypotheses (*Seeds et al., 2014*). That fruit flies may groom spontaneously in the absence of any apparent stimulus has also been previously suggested (*Connolly, 1968*; *Tinbergen, 1965*). Consistent with this, our work provides evidence that fruit flies groom as part of their daily repertoire of internally programmed behaviors and often without any obvious external stimulus. Our analysis revealed that over a period of hours, grooming is temporally structured by the fly circadian clock, with peak activity around dawn and dusk. The study also identifies transcription factors CLOCK and CYCLE as critical molecular components that control the amplitude of programmed *Drosophila* grooming.

Machine-learning is increasingly gaining popularity due to its applicability to virtually any problem involving pattern classification, including in studies aimed at deconstructing stereotyped behavior in the fruit fly (*Branson et al., 2009*; *Kabra et al., 2013*; *Kain et al., 2013*; *Mendes et al., 2013*; *Valletta et al., 2017*). Similar to these recent efforts, we constructed a computational pipeline incorporating elements of machine learning to automatically identify grooming events in video recordings of behaving flies. Our approach relies, in particular, on a supervised *k*-nearest neighbors algorithm to broadly classify behavior into grooming, locomotion and rest (*Figure 2*). Application of additional optional filters yielded approximate data on feeding and sleep (*Figure 4*). While previous methods offer important details on different modes of grooming (*Berman et al., 2014*; *Seeds et al., 2014*), leg movements (*Kain et al., 2013*; *Mendes et al., 2013*), and fly-fly interactions (*Branson et al., 2009*; *Kabra et al., 2013*) from short videos, the methods have limited capability for interpreting multi-day and multi-fly recordings. The method presented here offers less detail on modes of grooming, but can instead readily dissect circadian time-scale recordings into three to five behavioral classes on a typical personal computer.

The apparatus used in this method (*Figure 1*) also offers a number of advantages over current ones. First, most items used in the apparatus, including the ~6 cm tubes in which flies are visualized, are standard in a typical circadian experiment studying fly locomotion or sleep (*Lazopulo et al., 2015*; *Pfeiffenberger et al., 2010*) using the Drosophila Activity Monitor (DAM). The retention of this basic feature should lower the technical hurdle for the interested investigator who is likely to be one already engaged in locomotion and sleep studies in *Drosophila*. The use of a shared design to house flies also means that both experimental subjects and certain conclusions drawn from one platform may be readily transferred to the other. Most current grooming methods require specialized equipment for fly stimulation and detection (*Seeds et al., 2014*), elaborate optics (*Kain et al., 2013*), or a specific form of fluorescence microscopy (*Mendes et al., 2013*). Second, our apparatus can simultaneously monitor up to ~20 flies, while the existing approaches, although offering higher-resolution data, monitor only one animal at a time. The scalability and high-throughput nature of our platform should appeal to investigators interested in, for example, large-scale genetic studies to identify mechanisms that differentially affect grooming, locomotion and rest (*King et al., 2016*). Finally, the flies in our apparatus are allowed to move freely over a distance roughly 10 times their body length and still remain in the camera's field of view while technical constraints in other studies limit visualization to short distances (*Mendes et al., 2013*). The relative freedom of mobility, access to food, and long time-scales of observation offered by our apparatus thus facilitate analysis of basal, internally programmed behavior.

These properties make our platform amenable to addressing questions of biological relevance, such as the importance of grooming behavior, its temporal regulation with regards to other fly behaviors, and its dependence on the circadian timekeeping system. First, we found that flies

consistently devote a significant fraction of time to grooming behavior during periods of wakefulness (13%), and surprisingly, that grooming behavior is observed even during periods of reduced locomotor activity (*Figure 4—figure supplement 2A*). This suggests that the benefits of grooming outweigh the caloric resources expended and the resulting interruption of rest, underscoring the hypothesis that daily grooming is a fundamental behavior of *Drosophila*.

A few recent studies (*Hampel et al., 2015*; *Phillis et al., 1993*; *Seeds et al., 2014*) have shown that fly grooming can be directly induced by peripheral stimuli, and there has been considerable progress toward identifying the behavioral and neural aspects of such stimulus-induced grooming. However, programmed grooming, or grooming in the absence of a macroscopic stimulus, remains relatively understudied in *Drosophila*. To our knowledge, the existence of programmed grooming, first proposed in the mid 60s, still remains unreported.

Data from this study suggest that a significant portion of daily fly grooming is driven by internal programs. Flies in our experiments are active for ~34% of the time within a 24 hr period, during which they mostly engage in grooming, locomotion and feeding. Behavioral analysis showed that, like sleep, locomotion and feeding, fly grooming behavior is modulated by oscillations of the circadian clock (*Figure 5*). This finding raised the possibility that the observed grooming was stimulated by rhythms in contact with food or locomotor activity. However, closer examination revealed that kinetics in feeding and locomotion were distinct from those of grooming (*Figure 4—figure supplement 2*). Additionally, genetic modifications resulted in contrasting changes in these behaviors (*Figure 6*). These results together suggest that the majority of grooming events detected in our experiments are not triggered by external stimuli such as light, food and locomotor movements. Rather, internal regulatory mechanisms, independent of external stimuli, likely drive this programmed behavior.

Multi-day recordings of wild-type flies in constant darkness showed 24 hr rhythms in daily grooming patterns (*Figure 5*, *Figure 5—figure supplement 1*). Furthermore, these rhythms were shifted appropriately in the canonical period mutants $per^L$ and $per^S$ and abolished in arrhythmic $per^0$ flies (*Figure 5*). These data support a regulatory model in which timing of programmed grooming behavior is orchestrated by the circadian clock. Notably, since loss of rhythmicity did not significantly affect the amount of grooming (*Figure 6A*), our results suggest that the primary role of the clock is to organize the behavior in time without influencing the total time flies dedicate to grooming.

Intriguingly, two other circadian mutations, $cyc^{01}$ and $clk^{Jrk}$, increased the proportion of daily time flies spend grooming (*Figure 6B,C*), implying that the changes in grooming level may not be due to circadian defects. These data are consistent with the hypothesis that clock-independent but *cyc-* and *clk-* dependent pathways regulate the amount of programmed grooming behavior.

Finally, why are flies innately programmed to groom? The present study does not directly address this important question, but given that microscopic pathogens can sporulate on the fly cuticle and eventually infect the insect (*St. Leger et al., 2011*), persistent grooming may serve as a first line of defense against such attack. Thus, the immune system may constitute another internal program, similar to the *cyc* and *clk*-controlled mechanisms, that drives fly grooming; if so, we hypothesized that mutants with defective immune response may exhibit altered grooming behavior (*Lemaitre et al., 1995*; *Michel et al., 2001*). Consistent with this, we found that grooming was reduced in the immune-deficient *imd* mutant (*Figure 6—figure supplement 1H*), although a second immune-deficient strain lacking a member of the Toll pathway (*PGRP-SA^seml*) did not show a significant change. Further studies are required to clarify these initial results and elucidate the biological function of programmed grooming in *Drosophila*.

Together, our data provide strong supporting evidence for programmed grooming in *Drosophila* and suggest that this innate behavior is driven by two possibly distinct sets of regulatory systems. The circadian system temporally segregates time-dependent variations in grooming from those of other essential behavioral outputs like feeding and sleep. Circadian coordination of grooming underscores a previously under-appreciated importance of this behavior in the daily routine of the fruit fly. The second regulatory system adjusts the level of grooming relative to other behaviors. This set of regulation likely confers adaptability on the animal by allowing it to up- or downregulate grooming as necessitated by internal and external conditions. The dual control mechanism of grooming proposed here is highly reminiscent of the two-process framework—circadian and homeostatic—that is widely used in understanding sleep regulation (*Borbély, 1982*). Although this work has not

demonstrated grooming is under homeostatic control, future studies could be aimed at better characterizing the nature of the non-circadian regulatory system of fly grooming.

In summary, we present here a new platform to detect innate grooming behavior simultaneously and for days at a time in multiple individual fruit flies. The apparatus can be assembled easily, and the accompanying analytics are available publicly. Utilizing this platform, we report several mechanisms that are possibly responsible for driving the timing and level of programmed grooming in *Drosophila*. We also suggest future experiments that through use of this platform can lead to deeper understanding of the underlying biology of grooming and its relation to other essential fly behaviors.

# Materials and methods

## Key resources table

| Reagent type (species) or resource | Designation | Source or reference | Identifiers | Additional information |
|---|---|---|---|---|
| Strain, strain background (*Drosophila melanogaster*, male) | $sss^{P1}$ | DOI: 10.1126/science.1155942 | | on *iso31* background |
| Strain, strain background (*D. melanogaster*, male) | *iso31* | DOI: 10.1126/science.1155942 | | |
| Strain, strain background (*D. melanogaster*, male) | *fumin* | DOI: 10.1523/JNEUROSCI.2048-05.2005 | | on $w^{1118}$ background |
| Strain, strain background (*D. melanogaster*, male) | $w^{1118}$ | Bloomington Drosophila Stock Center | BDSC: 3605 | |
| Strain, strain background (*D. melanogaster*, male) | Canton S | Bloomington Drosophila Stock Center | BDSC: 64349 | |
| Strain, strain background (*D. melanogaster*, male) | $clk^{JRK}$ | this paper | | backcrossed for five generations to *iso31* |
| Strain, strain background (*D. melanogaster*, male) | $per^0$ | this paper | | backcrossed for five generations to $iso31^+$ |
| Strain, strain background (*D. melanogaster*, male) | $per^S$ | this paper | | backcrossed for six generations to $iso31^+$ |
| Strain, strain background (*D. melanogaster*, male) | $per^L$ | this paper | | backcrossed for six generations to $iso31^+$ |
| Strain, strain background (*D. melanogaster*, male) | $cyc^{01}$ | other | | on Canton S background, gifts from William Ja |
| Strain, strain background (*D. melanogaster*, male) | $iso31^+$ | other | | gifts from Michael Young |

## Fly strains

Clock mutants $per^S$, $per^L$, and $per^0$ were backcrossed for five-six generations to an *iso31* with *mini-white* insertion strain ($iso31^+$). $cyc^{01}$ flies, gifts from William Ja (The Scripps Research Institute), have the *Canton S* background. $Clk^{Jrk}$ flies were backcrossed for five generations to *iso31*. $sss^{P1}$ mutant flies, gifts from Amita Sehgal (Perelman School of Medicine at the University of Pennsylvania), have the *iso31* background. *fumin* mutants, gifts from F. Rob Jackson (Tufts University School of Medicine), have the $w^{1118}$ background. Flies were bred and raised at 23°C and 40% relative humidity on standard cornmeal and molasses food. All experiments were done with 5–8 days old males at $26^0$C and 70–80% relative humidity in a custom-built behavior tracking chamber (*Figure 1* and *Figure 2—*

*figure supplement 1A*). For each experiment, control strain refers to the genetic background of a mutant. WT flies in *Figures 4* and *5* refer to the *iso31⁺* line.

## Behavior tracking apparatus

### Chamber

Flies were placed individually in glass tubes (Trikinetics Inc., Waltham, MA, PGT5 × 65) with food and a cotton plug at opposite ends. Twenty tubes were placed on a custom-designed acrylic plate inside a transparent acrylic cuboid box for simultaneous imaging. Temperature and humidity were monitored every 5 min with a digital thermometer (Dallas Semiconductor, Dallas, TX, DS18B20) and a humidity sensor (Honeywell, Morris Plains, NJ, HIH-4010), respectively, while a wet sponge inside the chamber kept the relative humidity around 70–80% (*Figure 2—figure supplement 1A*).

### Illumination

The chamber was illuminated by two sets of light-emitting diode (LED) strips. White LEDs (LEDwholesalers, Hayward, CA, 2026) producing ~700 lux were used to simulate daytime conditions and infrared LEDs (LEDLIGHTSWORLD, Bellevue, WA, SMD5050-300-IR 850 nm) were used to visualize the flies at all times.

### Camera

A CCD monochrome camera (The Imaging Source, Charlotte, NC, DMK-23U445) fitted with a varifocal lens (Computar, Cary, NC, T2Z-3514-CS) was used for video imaging. To minimize influence of chamber's light/dark conditions on video quality, we put a 780 nm long pass filter (Midopt, Palatine, IL, LP780-30.5) in front of the lens. Videos were saved as 8-bit images in. avi format with 1280 × 960 resolution at 10 Hz and down-sampled as needed.

## Analytic hardware and runtime

Using a desktop computer with Intel Core i7-4770 3.4 GHz processer and 4 × 4 G DDR3 1600 MHz RAM, it takes ~7 hr to extract grooming, locomotion and rest data from an 8 hr video of 20 flies recorded in 10 Hz (in total 288,000 frames) at 1280 pixel ×960 pixel resolution. Videos are analyzed every two frames (5 Hz), which is sufficient to capture grooming events.

## Algorithm for automatic detection of grooming

All computational analyses were done with custom-written Matlab scripts that will be available at https://github.com/sbadvance/Drosophila-Grooming-Tracking.git (*Qiao, 2017;* copy archived at https://github.com/elifesciences-publications/Drosophila-Grooming-Tracking).

*Fly shape extraction.* Fly shape was extracted by applying a background subtraction algorithm. The background or reference frame is constructed randomly picking two frames, a template and a contrast, and comparing their pixel grayscale values and erasing all moving objects from the template frame. To remove the fly from the template frame, we replace the pixels belonging to the fly with corresponding pixels from contrast frames, relying on the fact that a fly is always darker than the surrounding objects. The template frame with no fly present then becomes the background frame. Additionally, because a fly's surroundings, including food debris, change substantially during the course of an experiment (*Figure 2—figure supplement 1B*), the background frame is regenerated every 1000 s. Lastly, if a fly occupies the same area in the template and contrast frames, the overlapping region cannot be erased on the template. To circumvent this problem, every time a background frame is generated, we randomly choose seven, instead of one, frames as contrast frames and compare all of them with the template. When a fly does not move for more than 1000 s, the fly will not be removed from the background and cannot be detected in other frames during this 1000 s. Thus, when a fly is not detected, we consider the fly to be stationary at the position where it was last detected.

To reduce effects of charge coupled device (CCD) image noise and fluctuations in the system, we set a minimum change $C_0$ as the threshold to accept grayscale changes from fly movements. We denote the grayscale value of a pixel located at (x, y) (in units of pixel, in our case, $x \in [1:1280]$, $y \in [1:960]$) in the template as $I_{template(x,y)}$ and in the contrast frame $I_{contrast(x,y)}$. Only if

$$I_{template(x,y)} - I_{contrast(x,y)} > C_0$$

then

$$I_{template(x,y)} = I_{contrast(x,y)}$$

While increasing threshold $C_0$ reduces noise, it can also lead to rejection of real movements of the fly. To optimize $C_0$, we tested noise levels in our images by analyzing a 3-hr video with dead flies. In the test, 30 pairs of consecutive frames were randomly chosen from the video and the differences between their corresponding grayscale pixel values were calculated. The distribution of the differences, stemming from noise, is shown in *Figure 2A*. Based on this distribution, we set $C_0$=10, which excludes 99.99% of noise-related changes in grayscale values.

*Feature normalization.* Since PM and CM both represent areas (number of pixels in area), while CD represents distance, we take the square root of PM and CM to make the dimensions of the features homogeneous. In addition, fly size varies between individuals and across experimental settings. To facilitate comparison of data in feature space, we therefore normalize PM, CM and CD of each fly with a scale parameter SP equal to the square root of the area of that fly. Thus, the final form of normalized features are

$$Normalized\ PM = \sqrt{PM}/SP$$

$$Normalized\ CM = \sqrt{CM}/SP$$

$$Normalized\ CD = CD/SP$$

## Spectral analysis

*Figures 4* and *5* and *Figure 5—figure supplements 1–3*: To measure periodicity in locomotion and grooming recordings, we applied the Lomb-Scargle periodogram (*Lazopulo et al., 2015*; *Scargle, 1982*) to time-series that were binned into 30 min periods. Power at indicated p values shown in power spectra were calculated according to

$$\mathrm{Power} = -\ln\left(1 - (1-p)^{1/N}\right)$$

where *p* is the *p*-value and *N* is the number of frequencies computed in Lomb-Scargle periodogram.

To test the effect of binning on rhythmicity, we binned grooming activity of individual flies in 30 min, 5 min, and 1 min bin sizes and ran Lomb-Scargle periodogram analysis on these binned data, as well as raw data without any additional binning. Examples of 5 individual spectra of each bin size are shown in *Figure 5—figure supplement 1C*. As shown in the figure, the separation between statistical cut-off power (at certain p value, horizontal lines) and peak power increases with smaller bin size or equivalently, larger number of data points (*N*). This is because in Lomb-Scargle periodogram, cut-off power grows as log (*N*) while peak power grows as *N*.

## Time series randomization

In *Figure 4F* and *Figure 5—figure supplement 2*, randomized grooming was generated by randomly shuffling time in raw grooming data. The corresponding modified locomotion and wake were calculated according to

Modified locomotion = original locomotion+original grooming – randomized grooming
Modified wake = original wakefulness+original grooming – randomized grooming
These manipulations modified either locomotion or wake while keeping the other unchanged.

## Statistics

No sample size estimation was performed when the study was being designed. Unless otherwise specified, quantitative experiments with statistical analysis were repeated at least three times independently. Exclusion of data applies to flies which were physically damaged (for example, broken wings or legs), physically confined (for example, trapped by condensation inside tubes), or dead during experiments. For testing statistical significance of differences between groups, we first tested

the normality of data by one-sample Kolmogorov-Smirnov test. Two-sample F-test was applied for equal variances test. Samples with equal variances were compared using two-tailed t-test. Satterthwaite's approximation for the effective degrees of freedom was applied for samples with unequal variances. Results were expressed as mean ± s.d., unless otherwise specified. *p<0.05, **p<0.01, ***p<0.001 were considered statistically significant.

In *Figure 4C,D* and *Figure 4—figure supplement 1B,C*, the Pearson correlation coefficient r for each pair of data was calculated according to the standard definition

$$r_{X,Y} = \frac{E[(X - \mu_X)(Y - \mu_Y)]}{\sigma_X \sigma_Y}$$

where *X* and *Y* are time spent in two behaviors X and Y, $r_{X,\ Y}$ is the Pearson correlation coefficient between two behaviors, $E[\ ]$ is the expectation value, $\mu$ and $\sigma$ are, respectively, mean value and standard deviation of a behavior. The statistical significance of r was estimated through bootstrapping. For each two behaviors, we randomly paired data from n flies (n = 84 for iso31+ and n = 76 for Canton S) and calculated a correlation coefficient r. This process was repeated 100,000 times and the empirical distribution of the randomly paired r values were used for a two-tailed test (*Figure 4—figure supplement 1D*). p-values for all Pearson correlation coefficients are presented in *Figure 4—figure supplement 1E*.

## Acknowledgements

This work was partially supported by the National Science Foundation under grant IOS-1656603 to SS and by National Institutes of Health grants R01GM105775 and R01AG045842 to MSH. The authors are grateful to William Ja, F Rob Jackson, Amita Sehgal and Michael Young for providing fly strains, Juan Lopez and Manuel Collazo for technical support and Stanislav Lazopulo and Andrey Lazopulo for suggestions and assistance with experiments. We thank Alan Li and Gadi Trocki for helpful comments on the manuscript.

## Additional information

### Funding

| Funder | Grant reference number | Author |
| --- | --- | --- |
| National Science Foundation | IOS-1656603 | Sheyum Syed |
| National Institutes of Health | R01GM105775 | Mimi  Shirasu-Hiza |
| National Institutes of Health | R01AG045842 | Mimi M Shirasu-Hiza |

The funders had no role in study design, data collection and interpretation, or the decision to submit the work for publication.

### Author contributions

Bing Qiao, Data curation, Software, Formal analysis, Validation, Investigation, Visualization, Writing—original draft; Chiyuan Li, Conceptualization, Software, Formal analysis, Methodology, Writing—original draft; Victoria W Allen, Investigation, Writing—review and editing; Mimi Shirasu-Hiza, Funding acquisition, Writing—review and editing; Sheyum Syed, Conceptualization, Supervision, Funding acquisition, Project administration, Writing—review and editing

### Author ORCIDs

Sheyum Syed http://orcid.org/0000-0002-4642-6678

### Decision letter and Author response

Decision letter https://doi.org/10.7554/eLife.34497.041
Author response https://doi.org/10.7554/eLife.34497.042

## Additional files

### Supplementary files
• Transparent reporting form
DOI: https://doi.org/10.7554/eLife.34497.035

### Major datasets
The following dataset was generated:

| Author(s) | Year | Dataset title | Dataset URL | Database, license, and accessibility information |
|---|---|---|---|---|
| Bing Qiao, Chiyuan Li, Victoria W Allen, Mimi Shirasu-Hiza, Sheyum Syed | 2018 | Automated analysis of internally programmed grooming behavior in *Drosophila* using a *k*-nearest neighbors classifier | https://doi.org/10.5061/dryad.94082 | Available at Dryad Digital Repository under a CC0 Public Domain Dedication |

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
