## [Decision Letter]

[Editors’ note: a previous version of this study was rejected after peer review, but the authors submitted for reconsideration. The first decision letter after peer review is shown below.]

Thank you for submitting your work entitled "Automated analysis of internally programmed grooming behavior in *Drosophila* using a *k*-nearest neighbors classifier" for consideration by *eLife*. Your article has been reviewed by three peer reviewers, and the evaluation has been overseen by a Reviewing Editor and a Senior Editor. The following individuals involved in review of your submission have agreed to reveal their identity: Benjamin de Bivort (Reviewer #1); Daniel Cavanaugh (Reviewer #3).

Our decision has been reached after consultation between the reviewers. Based on these discussions and the individual reviews below, we regret to inform you that your work will not be considered further for publication in *eLife*. Although all three reviewers found aspects of the work exciting, there were concerns about the manuscript as a methods paper and about the main biological finding that grooming is under circadian control preclude publication at this time.

Reviewer #1:

In this manuscript Qiao and Li et al. develop an automated annotation procedure for scoring grooming behavior in flies. They use this approach to then measure the circadian dependence of grooming, examining the effects of mutation in period, clock and cycle genes on grooming patterns. They find that the pattern of grooming is circadian and is disrupted by mutations that disrupt the circadian pattern of locomotion. They also examine the effect of stress on grooming. The major scientific result seems to be that control of grooming timing and duration are under independent genetic control, which I find plausible.

This work seems to be well done technically, and relatively clearly presented. The grooming detection method will be useful to other researchers. I am not sure of the significance of the scientific findings. It seems like it could be trivially explained by the circadian clock driving arousal, and once an animal is resolved it has to make an exclusive "choice" between grooming or locomoting. Thus, even if the circadian control of grooming is indirect, falling out of the direct control of locomotion by the circadian rhythm, the authors might see these results. Still maybe this counts as circadian control of grooming.

It doesn't feel like the paper is either clearly a methods paper or a circadian investigation. If it is the former, as suggested by the title, more work needs to be done to convey the generality and robustness of the classifier. If it's the latter, as suggested by the Abstract and the amount of Results section narrative devoted to the method (2p) vs circadian experiments (7.5), it seems fine but maybe in the wrong article category.

Lastly, as my expertise is not in circadian behavioral genetics, I can only say that their results in this area seem plausible and well performed but cannot evaluate them in the broader context of that subfield.

1) Were the flies used to assess the classifier accuracy all from the same genotype? If so which one, and do the authors have evidence that the classifier will generalize to other genotypes? Genotypes vary by size and pigmentation and this has the potential to change the value of the classification features and hence the classifier accuracy. Since the paper is given as a methods paper, more effort should be given to conveying the generality and robustness of the approach.

It should be easy to see if classification accuracy changes with the size of the segmented fly body from the data they have in hand, if there's enough natural variation in body size. But I don't think it's unreasonable to explore these questions with new measurements.

2) Subsection “Grooming plays an important role in the daily life of *Drosophila”* – Using distributions to assess behavioral stereotypy feels misguided, particularly as the authors have individual-level data. Individual flies have different behavioral biases in essentially all behaviors, so it's likely this is true for grooming. This could be true and nevertheless produce a normalized histogram that is relatively narrow. To assess, the authors could look at individual-level data of grooming rates across days. If every individual fly fills out the population distribution (i.e. the behavior is ergodic), then their argument would be supported.

The sentence in Subsection “Grooming plays an important role in the daily life of *Drosophila”* seems particularly speculative. More appropriate for the discussion? Even there, I don't see any reason why spontaneous behavior should be less idiosyncratic than stimulus evoked behavior. Perhaps it would even go the other way. All in all, this argument is not compelling.

The authors might want to, just for curiosity, look at the individual-level correlation between grooming and sleep.

3) Subsection “Temporal pattern of grooming is under control of the circadian clock” – The authors should clarify what they mean about testing whether grooming is under circadian control. In particular, if locomotion is given as under circadian control, and locomotion and grooming are definitionally exclusive, how can grooming *not be under circadian control? i.e. What is the result which would falsify the hypothesis that grooming is under circadian control?

As the authors go onto conclude in Subsection “Temporal pattern of grooming is under control of the circadian clock” that circadian systems regulate grooming it's particularly important to clarify what they mean early on.

The reported lag between peak locomotion and peak grooming of 2 hours as suggested by Figure 4 is not particularly convincing to me that these processes aren't trivially coupled in the sense of being both driven by arousal but exclusive of each other. Such an arrangement would be consistent with circadian rhythms driving arousal, and when an aroused animal doesn't walk it grooms. This doesn't feel like the kind of circadian control of grooming the authors are interested in, but they should clarify if this sort of arrangement passes the test.

Also, the data presented for a lag between grooming and running aren't particularly convincing as they appear to only happen 2/4 times. The sample size should be shown in Figure 4. Better would be to show the data points themselves.

Reviewer #2:

The authors develop methods to automate the detection of grooming (versus rest and locomotion) in *Drosophila*. They do so by using a trained classifier that relies on k-nearest neighbors clustering, to detect whether the fly periphery moves more than the core (grooming), the periphery and core move similarly (locomotion) or whether neither move much at all (rest). This is all performed on data from flies moving in small tubes, similar to those used extensively for circadian research. The authors manually label some data to assess the false positive and negative rates of their automated method – the algorithm is simple but nonetheless performs robustly with low false positive and negative rates. They then go on to collect data on fly grooming and locomotion over days and report on the statistics of these two behaviors over time, and to investigate whether grooming is modulated by circadian cycle. To do so, they use constant darkness and mutations in the period gene and find that they alter the rhythm of grooming but not the total time spent grooming each day. Finally, they examine mutations in cycle and clock genes, along with starvation conditions, and their effect on grooming rhythms, taking care to account for how flies change locomotor and feeding behavior concomitantly. This leads to the conclusion that there are two separate mechanisms that shape the rhythms of grooming (when flies groom throughout the day) and the duration of grooming (how long they spend grooming). I find the paper overall compelling for establishing a pipeline for studying fly grooming over long timescales – this setup should facilitate studies on the genetic and neural mechanisms underlying the regulation of grooming (versus locomotion, feeding, and sleep) over the course of days. However, there are a number of issues that I feel preclude publication of the manuscript in its current state.

1) After reading the title I expected a methods-heavy paper – however, much of the details on the automated grooming detector are in the Materials and methods section – and there is scant information describing the choices the authors made in how they extracted fly shape or optimized the classifier. Additionally, while the algorithm seems useful for studying grooming behaviors in these particular tubes under IR light, it is not clear how the algorithm would perform if the flies were in other (more natural) environments. The algorithm itself does not seem to be sufficiently innovative on its own to be of general interest, so it seems important for the authors to demonstrate its ability to detect grooming behaviors robustly in a number of conditions (thus making the algorithm useful to a number of investigators). On a related note, the statistics on grooming and locomotor behaviors reported in Figure 3 are likely to be dependent on the particular constraints of the environment these flies are in (housed singly, in thin narrow tubes, etc.) in addition to the genotypes of the flies (for example, Canton S, a lab inbred strain)- the authors should be clear about this in the text. In a different environment or with a different genetic background, the relative statistics of the two behaviors might be quite different.

2) The authors bin the locomotor and grooming data into ~3 minute bins, and then investigate rhythmicity using the Lomb-Scargle periodogram (a robust method for identifying rhythms in sparsely sampled data). This corresponds to a sampling rate of 0.006Hz – thus pushing any high frequency power in the grooming or locomotor rhythm down into the low frequencies (Figure 4), and it is not clear if the peaks in the power spectrum are significant. The authors should report on the statistical significance of the periodicity (if it exists) and also report on the effect of binning on the rhythm. Because of this issue, it is not clear if and how the period mutants affect the grooming patterns – grooming behavior may fluctuate over the course of the day, but the question the authors have not yet established is whether it is rhythmic.

3) A central claim of the paper is that grooming is not simply a correlate of locomotion/activity but is under separate control by the circadian clock, and a model is used (Figure 2) to show that grooming and locomotion are distinctly patterned during the day. However, the parameters are only useful if the model provides a good description of the data. How well does the model fit the data? Example fits, and quantification of fit quality should be provided. Related to this issue, differences in the temporal patterning of locomotion vs grooming only become apparent in the averaged and normalized data shown in the supplement (Figure S3A). But the normalization could introduce artifacts. For instance, the absolute basal rates for both behaviors appear similar in the raw data (Figure 3). After normalization, basal grooming rates appear elevated (Figure S3A) – but this is simply an effect of normalization to the max. The authors should ensure that this does not introduce artifacts in their model parameters.

4) The reduced variability of the fraction of time spent grooming vs. locomotion (Figure 3) could result from the fact that the mean and variance of the underlying variables are correlated, as is common for Bernoulli or Poisson processes. For example, the average rates for locomotion are higher than those for grooming – under a Poisson model, their standard deviation is expected to be higher as well. This trivial explanation should be ruled out, for example by looking at the correlation between individual fly means and standard deviations for locomotion alone (or grooming alone).

Reviewer #3:

The authors of this manuscript have made a significant advance in allowing for automated assessment of grooming behavior over long periods of time, and this will be generally useful to *Drosophila* researchers. Particularly beneficial is the fact that their system has the ability to monitor basal grooming, as opposed to induced grooming, which should be very informative to the field. The authors have also done a nice job of explaining in detail how their algorithm works, which appears to accurately detect grooming events.

There are several major problems, however, with the specific application of this system and the resulting conclusion that grooming is under circadian control and that the amount of grooming is regulated by the clock and cycle genes. Most concerning is that the authors have failed to use standard assessments, such as periodogram analysis of individual fly grooming behavior, to confirm a circadian pattern of grooming and to determine the robustness of the rhythm. The only quantification of free-running grooming rhythms is Lomb-Scargle periodogram analysis on mean grooming data over 4 days in constant darkness. We therefore have no idea as to the periods and powers of behavior of individual flies. The raw data traces shown suggest very weak rhythms. They should also show how clock and cycle mutations affect the circadian pattern of grooming-it's curious that they only show total amount of grooming for these mutants when a loss of rhythmicity would bolster the idea that grooming behavior is under circadian control.

In addition, the locomotor behavior of wildtype flies as determined by this monitoring system is unusual as it appears that a substantial amount (more than half) of activity is occurring during the dark period (see Figure 3—Figure supplement 1C). This calls into question either the accuracy of the algorithm to determine locomotion, the lighting and noise controls during the experiment, or the genetic background of the wildtype flies used.

Finally, it is not clear that the rhythms of grooming behavior are under direct circadian control as opposed to being a secondary product of sleep-wake cycles. Grooming can only occur when the flies are awake, which makes it very difficult to disentangle from sleep-wake rhythms. The authors have attempted to address this question by comparing the onset of grooming and locomotion during the evening (Figure 4), but locomotion is not the same thing as wakefulness, and thus it is possible that flies groom upon awakening from the afternoon siesta prior to engaging in large-scale movements. The authors should more closely analyze individual fly behavior to determine the temporal relationship between wakefulness and grooming. Unfortunately, short of identifying a mutant that selectively alters sleep-wake or grooming rhythms without affecting the other, it will be impossible to unequivocally address this concern.

The relationship between wakefulness and grooming might also explain why clkjrk and cyc01 mutants have increased grooming, as both mutants have decreased sleep overall (Hendricks et al., 2003; although note that cyc01 mutants used in this manuscript don't have reduced sleep amount as determined by the algorithm). It would be useful to look at grooming behavior in other short-sleeping mutants to see if it is similarly increased.

Overall, this manuscript represents a nice innovation that will benefit other researchers but has fallen short in using this innovation to uncover novel mechanisms regulating grooming behavior. At a minimum, additional analysis is needed to convincingly demonstrate circadian control of grooming.

[Editors’ note: what now follows is the decision letter after the authors submitted for further consideration.]

Thank you for resubmitting your work entitled "Automated analysis of internally programmed grooming behavior in *Drosophila* using a k-nearest neighbors classifier" for further consideration at *eLife*. Your revised article has been favorably evaluated by K VijayRaghavan (Senior editor), Kristin Scott (Reviewing editor), and three reviewers.

The manuscript has been improved but there are some remaining issues that need to be addressed as text changes before acceptance, as outlined below:

1) Independence of grooming cycles from locomotion/wakefulness cycles is challenging to show definitively. At this stage, the authors have provided evidence that grooming is not *trivially* rhythmic because of wakefulness rhythmicity (i.e. there's not a wake/sleep cycle and 1/nth of the time you're awake you groom). Clearly these are coupled processes in that they are hierarchical – you have to be awake to groom. The authors should restate their discussion to say that they haven't definitively demonstrated that grooming rhythms are independent of the sleep-wake cycle.

2) Separability of the modes of regulating grooming is not so clear, since the genes that seem to be require for the separate modes are all part of the core clock machinery. The justification for the idea of two separate programs in the current paper rests primarily on the fact that mutations in the per gene result in loss of grooming rhythms but not grooming amount, while mutations in clock and cyc affect both the rhythm and amount of grooming. The main problem here is that clock and cyc are core elements of the clock, and thus can't be separated from their role in driving rhythmicity. Furthermore, per is part of the negative arm of the clock while clock and cyc are part of the positive arm. Couldn't it be that the positive components, when missing, lead to increased grooming, but the negative components do not? The data do not prove that there are separate programs controlling the timing and amount of grooming. Without that additional evidence, this conclusion be toned down.

3) To address concerns about binning their data, the authors now test 3 different bin sizes (1, 5, and 30 minute bins) and then search for low frequency rhythms – this seems okay, since for all cases the bin width is much smaller than the underlying rhythm they are trying to detect (24 hour circadian cycle). However, the authors should be able to run the Lomb-Scargle analysis on the data directly, without binning, and still uncover the rhythms, no? If so, this argues against a need for binning at all.

4) In subsection “Automatic grooming detecting system”, the authors say: "we used a system that incorporates features from *Drosophila* Activity Monitors (DAMs) with a custom video set-up". Based on my understanding, the only feature of the DAM system that the authors have incorporated is the 5mm diameter glass tubes to house the flies. It seems misleading to say that they are using features of the DAM system, which would imply use of the monitoring technology (i.e. the actual monitor itself).

5) In Figure 2, the word "periphery" is misspelled.

6) There is no mention of the Pearson correlation test in the methods section. I would recommend that the authors include p values in addition to the correlation coefficient. In Figure 4—figure supplement 1, it would be nice for the order of the Pearson correlation graphs to match the order from Figure 4.

7) Are the data from Figure 5—figure supplement 2 collected in DD conditions?

8) This is a stylistic consideration, but some of the new figure captions (for example the caption for Figure 6), read like a Results section. They don't clearly explain what is being graphed and instead make conclusions about the data.

9) There are some residual typos, such as missing spaces before references and sporadic double spaces. There are almost certainly other typos that I failed to see in reading it. So careful copy editing should be done to reduce the number of these that make it into print.

10) The authors might want to consider versions of the title along these lines: "Automated classification of grooming behavior in *Drosophila* reveals independent modes of genetic control " This reflects a few suggestions: (1) I don't think it's essential to mention kNN in the title. Other clustering approaches would presumably work in that space, (2) "internally programmed behavior" still strikes me as an odd framing. I'd rather see a quick summary of their major science finding. Those are my two cents. Happy to leave this up to the authors.

---

## [Author Response]

[Editors’ note: the author responses to the first round of peer review follow.]

Reviewer #1:1) Were the flies used to assess the classifier accuracy all from the same genotype? If so which one, and do the authors have evidence that the classifier will generalize to other genotypes? Genotypes vary by size and pigmentation and this has the potential to change the value of the classification features and hence the classifier accuracy. Since the paper is given as a methods paper, more effort should be given to conveying the generality and robustness of the approach.It should be easy to see if classification accuracy changes with the size of the segmented fly body from the data they have in hand, if there's enough natural variation in body size. But I don't think it's unreasonable to explore these questions with new measurements.

This is an excellent point. In the previous version, we overlooked mentioning the genotype. The genotype is now explicit in the manuscript. All training samples shown in Figure 2 are from *w1118* flies (N=20). Since individual body sizes vary even within the same genotype, we created the PM, CD and CM feature space (Figure 2) in terms of the features normalized by individual fly body size, as described in Materials and methods section.

The reviewer is correct – pigmentation and other morphological differences can also potentially affect classifier accuracy. To investigate the influence of these factors, we performed additional analyses assessing the accuracy metrics of the *w1118*–based training set for individuals from three other commonly used laboratory strains – *Canton S, iso31*, and *yw*. Each of the three videos was 10-minutes long and included 20 flies. As shown in Figure 3, error rates in all tested strains are less than 10%. Details of these studies and their findings are now included in the manuscript. The text now reads (subsection “Behavior classification algorithm”):

“Since size and pigmentation differences between genotypes can potentially affect behavioral classification, we investigated robustness of our *w1118*-trained classifier with manually-labeled data from *Canton S, iso31*, and *yw* strains (10-minute videos with N=20 of each type). As shown in Figure 3, error rates in all tested strains are less than 10%. Together, these results suggest that our method identifies grooming with high fidelity in several different *Drosophila melanogaster* strains.”

2) Subsection “Grooming plays an important role in the daily life of Drosophila” – Using distributions to assess behavioral stereotypy feels misguided, particularly as the authors have individual-level data. Individual flies have different behavioral biases in essentially all behaviors, so it's likely this is true for grooming. This could be true and nevertheless produce a normalized histogram that is relatively narrow. To assess, the authors could look at individual-level data of grooming rates across days. If every individual fly fills out the population distribution (i.e. the behavior is ergodic), then their argument would be supported.

We thank the reviewer for raising concerns about the presentation. We should not have presented the data using a normal distribution as it clearly conveyed the wrong point and was not appropriate for the purpose. Our intention was simply to draw attention to the observation that grooming has a smaller population-wide variance. In the revised version, because of the new structure of our work, this comparison between locomotion and grooming is removed from the manuscript. In Author response image 1 we show the confusing presentation and the simple graph that conveys our point.

In the panel on right, inter-individual differences in daily grooming and locomotion are presented. Each point is standardized daily grooming or locomotion of an individual fly, which is calculated by dividing daily among of grooming(green) or locomotion(gray) of an individual fly by the respective population average. The variation in standardized daily grooming time among individuals (coefficient of variation) is significantly less than in locomotion. The coefficient of variation of grooming is 0.14 compared with 0.34 for locomotion. N=66 flies. In the previous version (left panel), standardized daily grooming or locomotion of individual flies were fitted to normal distributions.

The sentence in Subsection “Grooming plays an important role in the daily life of Drosophila” seems particularly speculative. More appropriate for the discussion? Even there, I don't see any reason why spontaneous behavior should be less idiosyncratic than stimulus evoked behavior. Perhaps it would even go the other way. All in all, this argument is not compelling.The authors might want to, just for curiosity, look at the individual-level correlation between grooming and sleep.

We have removed the comparison of population-wide variance between grooming and locomotion in the revised manuscript, as mentioned above. Thus, this sentence is no longer there.

We thank the reviewer for the advice to look at individual-level correlations. We have added individual-level correlation among grooming, locomotion, feeding, short rest, and sleep of wt to Figure 4 (*iso31+*) and in Figure 4—figure supplement 1 (*Canton S*). In addition, correlation between grooming and locomotion and between grooming and sleep in individual sleep mutants *sss* and *fumin*, and circadian mutants *per^0^, clk^JRK^* and *cyc^01^*flies are provided in Figure 6—figure supplement 1.

3) Subsection “Temporal pattern of grooming is under control of the circadian clock” – The authors should clarify what they mean about testing whether grooming is under circadian control. In particular, if locomotion is given as under circadian control, and locomotion and grooming are definitionally exclusive, how can grooming *not be under circadian control? i.e. What is the result which would falsify the hypothesis that grooming is under circadian control?As the authors go onto conclude in Subsection “Temporal pattern of grooming is under control of the circadian clock” that circadian systems regulate grooming it's particularly important to clarify what they mean early on.

The issue of circadian control of grooming is an important result of this work and we thank the reviewer for pointing out insufficient clarity in the definitions and test criteria. Previous wording may have incorrectly implied that locomotion and grooming are always mutually exclusive. They are mutually exclusive at the level of individual events but not at the level of fractional duration in each behavior. The latter quantity is what we show in time-series and where any long-term oscillations would appear. Within the state of wakefulness, a fly can locomote, groom, feeding or rest. As a result, wakefulness can be rhythmic as long as only one of the four is rhythmic. Therefore, if we find locomotion to be rhythmic, there is no requirement on either grooming or rest to be rhythmic. To test the presence of circadian rhythmicity in grooming behavior, it was therefore necessary to monitor the behavior in constant environment (without any external light or temperature cues) and check for its rhythmicity.

Much of this reasoning was missing in the previous version. But, prompted by the referee’s questions, we have now added substantial additional text starting in the last paragraph of subsection “Flies spend a significant portion of their awake time grooming”. We have also redesigned figures and added new figures to address this important question: Figure 4, Figure 4—figure supplement 1 and Figure 5—figure supplement 2.

The reported lag between peak locomotion and peak grooming of 2 hours as suggested by Figure 4 is not particularly convincing to me that these processes aren't trivially coupled in the sense of being both driven by arousal but exclusive of each other. Such an arrangement would be consistent with circadian rhythms driving arousal, and when an aroused animal doesn't walk it grooms. This doesn't feel like the kind of circadian control of grooming the authors are interested in, but they should clarify if this sort of arrangement passes the test.Also, the data presented for a lag between grooming and running aren't particularly convincing as they appear to only happen 2/4 times. The sample size should be shown in Figure 4. Better would be to show the data points themselves.

We updated number of samples and now this figure has been moved to Figure 4—figure supplement 2 to make room for new panels in main figure. Sample size was N= 50 *iso31+* flies.

Reviewer #2:1) After reading the title I expected a methods-heavy paper – however, much of the details on the automated grooming detector are in the Materials and methods section – and there is scant information describing the choices the authors made in how they extracted fly shape or optimized the classifier. Additionally, while the algorithm seems useful for studying grooming behaviors in these particular tubes under IR light, it is not clear how the algorithm would perform if the flies were in other (more natural) environments. The algorithm itself does not seem to be sufficiently innovative on its own to be of general interest, so it seems important for the authors to demonstrate its ability to detect grooming behaviors robustly in a number of conditions (thus making the algorithm useful to a number of investigators). On a related note, the statistics on grooming and locomotor behaviors reported in Figure 3 are likely to be dependent on the particular constraints of the environment these flies are in (housed singly, in thin narrow tubes, etc.) in addition to the genotypes of the flies (for example, Canton S, a lab inbred strain)- the authors should be clear about this in the text. In a different environment or with a different genetic background, the relative statistics of the two behaviors might be quite different.

a) We agree with the reviewer that more information about our approach ought to be included outside of Methods and in the main body of the manuscript. We have added more details on fly shape extraction, features extraction, model optimization, and validation in the Results section. We have put more data on pruning filter size optimization in Figure 3 and accuracy tests of our methods on different wt strains in Figure 3.

b) This study intended to devise a platform that would add high-throughput detection of fly grooming to the current list of fly circadian behaviors studied in the laboratory. Since locomotor activity and sleep are two of the most commonly studied circadian behaviors and the studies use the same experimental design (*Drosophila* Activity Monitor (DAM)), we attempted to retain as much of that design as possible in designing the new platform. The choice of ~6 cm long glass tubes to house individual animals retains an important feature of ongoing fly circadian experiments and lowers the technical hurdle of setting up the platform for the interested researcher (who is more likely to be one already studying locomotion and/or sleep). This also means the same fly preliminarily measured using DAM can be promptly placed in the new platform without the need for any special handling/transfer steps. Also, a shared arena design means that conclusions from one platform on some aspects of locomotion and sleep can be assumed to be valid in another, thus speeding up progress.

Although we have not conducted a comprehensive study, limited tests suggest our method should work well in other environments as long as flies are solitary and back-lit with a stable infra-red source.

c) Thank you for bringing up this important point. We agree that some of the behavioral statistics are potentially specific to the environment in which the flies are housed. Some features are clearly dependent on the genotype as well, as revealed by our data (e.g. Figure 4 and Figure 4—figure supplement 1 A-C; Figure 6 and Figure 6—supplement 1). We now further stress these points in subsection “Flies spend a significant portion of their awake time grooming” with additional references:

“It is worth noting here that such behavioral statistics can vary even between wild-type laboratory strains (Colomb et al., 2015; Zalucki et al., 2015)

2) The authors bin the locomotor and grooming data into ~3 minute bins, and then investigate rhythmicity using the Lomb-Scargle periodogram (a robust method for identifying rhythms in sparsely sampled data). This corresponds to a sampling rate of 0.006Hz – thus pushing any high frequency power in the grooming or locomotor rhythm down into the low frequencies (Figure 4), and it is not clear if the peaks in the power spectrum are significant. The authors should report on the statistical significance of the periodicity (if it exists) and also report on the effect of binning on the rhythm. Because of this issue, it is not clear if and how the period mutants affect the grooming patterns – grooming behavior may fluctuate over the course of the day, but the question the authors have not yet established is whether it is rhythmic.

We apologize for not reporting significance metrics for the circadian peaks in the original manuscript. We now include a few example power spectra (with p values) and detailed statistics of all tested control and mutants in Figure 5. Accompanying text in Results has been updated. A new supplementary figure, Figure 5—figure supplement 1, has been added to show additional example power spectra (with p=0.05 and p=0.01 cut offs) of wild-type and mutant flies. These details clearly show that grooming rhythms are statistically significant in wild-type and period-shifted mutants and well-below the cut-off in the classic arrhythmic flies.

To test the effect of binning on rhythmicity, we binned grooming activity of individual flies in 30minutes, 5-minutes, and 1-minute bin sizes and ran Lomb-Scargle periodogram analysis on these binned data. Examples of 5 individual spectra of each bin size are shown in Figure 5—figure supplement 1. As shown in the figure, the separation between statistical cut-off power (at certain p value, horizontal lines) and peak power increases with smaller bin size or equivalently, larger number of data points (*N*). This is because in Lomb-Scargle periodogram, cut-off power grows as ln*(N)* while peak power grows as *N*. This detail has been added to Methods section.

3) A central claim of the paper is that grooming is not simply a correlate of locomotion/activity but is under separate control by the circadian clock, and a model is used (Figure 2) to show that grooming and locomotion are distinctly patterned during the day. However, the parameters are only useful if the model provides a good description of the data. How well does the model fit the data? Example fits, and quantification of fit quality should be provided.

We thank the reviewer for pointing out this missing important detail. A summary about the model and method, including analytic functions used in the fitting procedure and example fits, is now in Figure 4—figure supplement 4. In addition, fitting errors are shown in the accompanying Table 1 and Table 2 in Figure 4—figure supplement 3.

Related to this issue, differences in the temporal patterning of locomotion vs grooming only become apparent in the averaged and normalized data shown in the supplement (Figure S3A). But the normalization could introduce artifacts. For instance, the absolute basal rates for both behaviors appear similar in the raw data (Figure 3). After normalization, basal grooming rates appear elevated (Figure S3A) – but this is simply an effect of normalization to the max. The authors should ensure that this does not introduce artifacts in their model parameters.

The actual fittings of the data were done on the raw activity and power spectrum of individual flies, without any normalization. It is true that the absolute value of basal rate of grooming is lower than that of locomotion (previous Figure 3). The point for normalization in Figure S3A (removed in revised version) was to allow easy comparison of the two behaviors and show the smaller relative day-night difference (smaller amplitude) in grooming than in locomotion.

4) The reduced variability of the fraction of time spent grooming vs. locomotion (Figure 3) could result from the fact that the mean and variance of the underlying variables are correlated, as is common for Bernoulli or Poisson processes. For example, the average rates for locomotion are higher than those for grooming – under a Poisson model, their standard deviation is expected to be higher as well. This trivial explanation should be ruled out, for example by looking at the correlation between individual fly means and standard deviations for locomotion alone (or grooming alone).

We thank the referee for this advice. Substantial revision and streamlining precluded, this comparison between locomotion and grooming from the current manuscript.

But regarding the now excluded figure: It is true that variance and mean of grooming and locomotion of individual flies could be correlated. In the previous Figure 3 we presented the variability of daily grooming and locomotion among individuals in the population. In the figure, daily grooming or locomotion of individual flies was standardized by dividing by the population average grooming or locomotion (coefficients of variation of grooming and locomotion). Based on the central limit theorem, the distribution of both standardized behaviors should be normally distributed. Thus, in the two distributions that were presented, mean and variance should not be correlated.

Reviewer #3:There are several major problems, however, with the specific application of this system and the resulting conclusion that grooming is under circadian control and that the amount of grooming is regulated by the clock and cycle genes. Most concerning is that the authors have failed to use standard assessments, such as periodogram analysis of individual fly grooming behavior, to confirm a circadian pattern of grooming and to determine the robustness of the rhythm. The only quantification of free-running grooming rhythms is Lomb-Scargle periodogram analysis on mean grooming data over 4 days in constant darkness. We therefore have no idea as to the periods and powers of behavior of individual flies. The raw data traces shown suggest very weak rhythms. They should also show how clock and cycle mutations affect the circadian pattern of grooming-it's curious that they only show total amount of grooming for these mutants when a loss of rhythmicity would bolster the idea that grooming behavior is under circadian control.

We apologize for this important omission. We have updated Figure 5, Figure 5—figure supplement 1, and Figure 5—figure supplement 3 with examples of individual power spectra and detailed circadian-rhythm related statistics of wild-type, per mutants, *clk^JRK^* and *cyc^01^*flies. All single fly power spectra now appear with power at the p=0.05 and p=0.01 levels (horizontal dashed and dash dot lines) indicated.

These details now offer clearer picture of circadian rhythmicity in grooming of wild-type and period-shifted mutants and their absence in canonical arrhythmic mutants.

In addition, the locomotor behavior of wildtype flies as determined by this monitoring system is unusual as it appears that a substantial amount (more than half) of activity is occurring during the dark period (see Figure 3—Figure supplement 1C). This calls into question either the accuracy of the algorithm to determine locomotion, the lighting and noise controls during the experiment, or the genetic background of the wildtype flies used.

All per mutant strains we used in this work are backcrossed to an iso31 line with miniwhite insertion (*iso31+*). Based on raw data from Figure 3—figure supplement 1C (now Figure 5—figure supplement 3 in revised version), in 12-12 light-dark experiments these wildtype flies do show higher level of locomotion activity during night (~41%) than during day (~32%)The ratio of nightime to day-time locomotor activity is ~1.28. To validate data from our new platform, we tracked locomotor behavior of the *iso31+* flies with the traditional single IR beam monitors in a commercial incubator with superior light and noise control than the enclosure for grooming experiments. Note activity measurement units are different from the two systems – video data yield time spent in locomotion and IR measurements yield beam breaks per unit time – and are therefore not directly comparable. Examples of single beam LD data of individual flies, Author response image 2, show more locomotion activity during dark period than during light, the ratio being 1.11, consistent with our video tracking data. The comparison rules out the possibilities that night-time activity is caused by algorithmic error or experimental noise.

**Author response image 2. respfig2:** 

Finally, it is not clear that the rhythms of grooming behavior are under direct circadian control as opposed to being a secondary product of sleep-wake cycles. Grooming can only occur when the flies are awake, which makes it very difficult to disentangle from sleep-wake rhythms. The authors have attempted to address this question by comparing the onset of grooming and locomotion during the evening (Figure 4), but locomotion is not the same thing as wakefulness, and thus it is possible that flies groom upon awakening from the afternoon siesta prior to engaging in large-scale movements. The authors should more closely analyze individual fly behavior to determine the temporal relationship between wakefulness and grooming. Unfortunately, short of identifying a mutant that selectively alters sleep-wake or grooming rhythms without affecting the other, it will be impossible to unequivocally address this concern.

We thank the reviewer for pointing out these subtleties in the relationship between the various behaviors and what they mean for rhythmicity of grooming. Our interpretation of temporal constraints on the behaviors was missing in the previous version and that may have been responsible for the perceived incorrect implication that locomotion and grooming are mutually exclusive in terms of fractional time spent in the two behaviors. At any point in time, flies in our experiments can transition into locomotion, grooming, feeding, short rest, or sleep. Although individual events of these different states are mutually exclusive, once they are binned in time, the behaviors lose their rigid mutual exclusivity. This means that within a given time bin (as in the fractional time data traces) a fly can be engaged in multiple behaviors, as long as the first four (constituting wakefulness) and sleep add up to 1 ().Figure 4—figure supplement 1 Complementary relationship between wakefulness and sleep imply that if one is rhythmic, the other must also be rhythmic. However, a requirement of rhythmic wakefulness does not necessitate every one of the four constituent behaviors to be rhythmic but rather only one to be rhythmic (preferably one in which flies spend the most time). Since locomotion is already widely considered to be rhythmic, there was no prior mathematical burden on grooming, feeding or rest to vary rhythmically. Therefore, to test the presence of circadian rhythmicity in grooming behavior, it was necessary to monitor the behavior in constant darkness and check for its rhythmicity.

Much of this reasoning was missing in the previous version. But, prompted by the referee’s question, we have now added additional text (subsection “Flies spend a significant portion of their awake time grooming”), Figure 4, and Figure 5—figure supplement 2.

Previous Figure panel 4E has been moved to Figure 4—figure supplement 2 in order to make room for additional panels in Figure 4. The purpose of that panel was to simply demonstrate differences in circadian regulation of grooming vs locomotion and grooming vs feeding within the state of wakefulness.

The relationship between wakefulness and grooming might also explain why clkjrk and cyc01 mutants have increased grooming, as both mutants have decreased sleep overall (Hendricks et al., 2003; although note that cyc01 mutants used in this manuscript don't have reduced sleep amount as determined by the algorithm). It would be useful to look at grooming behavior in other short-sleeping mutants to see if it is similarly increased.

Our *clk^JRK^* flies groom much more but sleep significantly less than their controls (*clk^JRK^* groom 9% and sleep 32% as opposed to control grooming at 6% and sleep at 56%). In contrast, *cyc^01^* flies groom more compared to their control strain (16% vs. 9%) but sleep about the same (51% vs. 52%). This indicates that increased grooming is not necessarily a result of increased wakefulness (or decreased sleep).

Following the referee’s suggestion, we looked at grooming behavior in other short sleep mutants, *fumin* and *sleepless.* Compared with wt flies, sleep time in *fumin* flies decreases significantly, while daily grooming increases only ~1.7% (Figure 6). On the other hand, *sleepless* flies show significant decrease in both behaviors (Figure 6). These trends are also visualized now in Figure 6—figure supplement 1.

These data again suggest that grooming is not simply correlated with wakefulness.

[Editors' note: the author responses to the re-review follow.]

The manuscript has been improved but there are some remaining issues that need to be addressed as text changes before acceptance, as outlined below:1) Independence of grooming cycles from locomotion/wakefulness cycles is challenging to show definitively. At this stage, the authors have provided evidence that grooming is not *trivially* rhythmic because of wakefulness rhythmicity (i.e. there's not a wake/sleep cycle and 1/nth of the time you're awake you groom). Clearly these are coupled processes in that they are hierarchical – you have to be awake to groom. The authors should restate their discussion to say that they haven't definitively demonstrated that grooming rhythms are independent of the sleep-wake cycle.

We agree with the reviewer that a more thorough study is required to show complete independence of rhythms in grooming and wakefulness. We simply wish to make the reader aware that rhythmic wakefulness does not directly imply rhythmic grooming – a level of independence permitted within the hierarchical relationship between the two behaviors – only to motivate a need for grooming experiments in constant darkness. Independence in the opposite direction is not critical to our work. We have added the following statements to further alert the reader of the incompleteness of our results (subsection “Temporal pattern of grooming is controlled by the circadian clock”):

“It should be noted here that our simulation results do not demonstrate bidirectional independence of rhythmicity in wakefulness and grooming but, only that rhythmicity of wakefulness does not depend on that of grooming. Demonstration of fully independent rhythms in the two behaviors is beyond the scope of the present study.”

2) Separability of the modes of regulating grooming is not so clear, since the genes that seem to be require for the separate modes are all part of the core clock machinery. The justification for the idea of two separate programs in the current paper rests primarily on the fact that mutations in the per gene result in loss of grooming rhythms but not grooming amount, while mutations in clock and cyc affect both the rhythm and amount of grooming. The main problem here is that clock and cyc are core elements of the clock, and thus can't be separated from their role in driving rhythmicity. Furthermore, per is part of the negative arm of the clock while clock and cyc are part of the positive arm. Couldn't it be that the positive components, when missing, lead to increased grooming, but the negative components do not? The data do not prove that there are separate programs controlling the timing and amount of grooming. Without that additional evidence, this conclusion be toned down.

We agree with the reviewer that we have not shown conclusively that the two modes of grooming control are independent. The following changes have been made in response to this comment:

Abstract: “One of these programs regulates the timing of grooming and involves the core circadian clock components cycle, clock, and period. The second program regulates the duration of grooming and, while dependent on cycle and clock, appears to be independent of period.”

Subsection “Grooming duration is controlled by cycle and clock”: “Importantly, together with per^0^ data, the results raise the possibility of non-circadian roles for cyc and clk in setting the duration of internally driven grooming in *Drosophila*.”

Subsection “Grooming duration is controlled by cycle and clock”: “The apparent absence of per from the second regulatory mechanism is consistent with the possibility that the two control mechanisms are operate independently.”

Discussion section: “…implying that the changes in grooming level are not may not be due to circadian defects. These data are consistent with the hypothesis that clock-independent but cyc- and clk- dependent pathways regulate the amount of programmed grooming behavior”.

Discussion section: suggest that this innate behavior is driven by two possibly distinct sets of regulatory systems.

3) To address concerns about binning their data, the authors now test 3 different bin sizes (1, 5, and 30 minute bins) and then search for low frequency rhythms – this seems okay, since for all cases the bin width is much smaller than the underlying rhythm they are trying to detect (24 hour circadian cycle). However, the authors should be able to run the Lomb-Scargle analysis on the data directly, without binning, and still uncover the rhythms, no? If so, this argues against a need for binning at all.

We thank the reviewer for highlighting this point. The time series in Figure 4, Figure 5, and Figure 5—figure supplement 1, Figuer 5—figure supplement 2 and Figure 5—figure supplement 2 were binned to bring out long timescale patterns in the data. Display of the raw 5 Hz data would not demonstrate this point as effectively, as the raw data would be dominated by short timescale fluctuations. A 30-minute bin size was chosen as a balance between averaging over the short time fluctuations but without loss of the ~24 hr oscillations. As the reviewer correctly points out, periodogram analysis can be done on the raw data. We have now added an additional column in Figure 5—figure supplement 1 showing Lomb-Scargle analysis of the 5 Hz data. Compared to the binned data, difference in power between the circadian peak and the statistical cut-off is even larger for the 5 Hz data, in accordance with our statement in subsection “Spectral analysis”, that in power spectrum the peak power grows as *N* while the cut-off power grows as log *N*.

4) In subsection “Automatic grooming detecting system”, the authors say: "We used a system that incorporates features from Drosophila Activity Monitors (DAMs) with a custom video set-up". Based on my understanding, the only feature of the DAM system that the authors have incorporated is the 5mm diameter glass tubes to house the flies. It seems misleading to say that they are using features of the DAM system, which would imply use of the monitoring technology (i.e. the actual monitor itself).

We concur with the reviewer that the sentence may lead to confusion. The statement has been replaced with a new one that does not refer to the DAM system:

“We used a custom-designed video set-up to monitor fly behavior.”

5) In Figure 2, the word "periphery" is misspelled.

We thank the reviewer for pointing out the error. The spelling has been corrected.

6) There is no mention of the Pearson correlation test in the methods section. I would recommend that the authors include p values in addition to the correlation coefficient. In Figure 4—figure supplement 1, it would be nice for the order of the Pearson correlation graphs to match the order from Figure 4.

We apologize for not reporting significance metrics for the Pearson correlation coefficient in the manuscript. We now include p-values for Pearson coefficient in all correlation analyses. When we calculated the bivariate normality of pairs of data in each correlation graph, we noticed that some pairs deviated strongly from a normal distribution. Thus, instead of testing significance of r values with Student’s t-test, we applied the bootstrap method for calculation of p-values. We now include example empirical distribution of r values in Figure 4—figure supplement 1 and provide all p-values in Figure 4—figure supplement 1. Details of the test are described in Materials and methodssubsection “Statistics” as follows:

“In Figure 4 and Figure 4—figure supplement 1, the Pearson correlation coefficient r for each pair of data was calculated according to the standard definition

rX,Y=EX-μXY-μYσXσY

where X and Y are time spent in two behaviors X and Y, rX,Y is the Pearson correlation coefficient between two behaviors, E[] is the expectation value, μ and σ are, respectively, mean value and standard deviation of a behavior. The statistical significance of r was estimated through bootstrapping. For each two behaviors, we randomly paired data from n flies (n=84 for iso31+ and n=76 for Canton S) and calculated a correlation coefficient r. This process was repeated 100000 times and the empirical distribution of the randomly paired r values were used for a two-tailed test (Figure 4—figure supplement 1). p-values for all Pearson correlation coefficients are presented in Figure 4—figure supplement 1.”

Also, the order of the Pearson correlation graphs in Figure 4—figure supplement 1 is now matched with that in Figure 4.

7) Are the data from Figure 5—figure supplement 2 collected in DD conditions?

Yes, they are. We noticed that this information was before mentioned only in the main text. Now, it is specified in the figure caption as well.

“Time series in the four examples were taken in constant darkness (DD) and[…]”8) This is a stylistic consideration, but some of the new figure captions (for example the caption for Figure 6), read like a Results section. They don't clearly explain what is being graphed and instead make conclusions about the data.

We regret not having included details. We have added more information about the figure in caption for Figure 6 and now it reads:

“In each panel, bar plots on left show average fractional time spent in grooming in mutant and control flies. Pie charts on right present average fractional time spent in grooming (green), locomotion (gray), sleep (dark gray), short rest (purple) and feeding (blue). Here, numerical values for fractional time spent in behavior are indicated only for grooming, locomotion and sleep[…]”

9) There are some residual typos, such as missing spaces before references and sporadic double spaces. There are almost certainly other typos that I failed to see in reading it. So careful copy editing should be done to reduce the number of these that make it into print.

We thank the reviewer for this comment and have now corrected 14 similar typos.

10) The authors might want to consider versions of the title along these lines: "Automated classification of grooming behavior in Drosophila reveals independent modes of genetic control " This reflects a few suggestions: (1) I don't think it's essential to mention kNN in the title. Other clustering approaches would presumably work in that space, (2) "internally programmed behavior" still strikes me as an odd framing. I'd rather see a quick summary of their major science finding. Those are my two cents. Happy to leave this up to the authors.

We appreciate the referee’s suggestion. Considering (1) this is a methods/technique paper (2) referee comment above about further ‘toning down’ of the grooming control model we propose, and (3) the suggestion to replace ‘internally programmed’ from title, we decided to keep ‘kNN’, not mention ‘independent control’ and replace ‘internally programmed’ with ‘long-term’ in the title. The new title is:

“Automated analysis of long-term grooming behavior in *Drosophila* using a k-nearest neighbors classifier”